# Inositol pyrophosphates promote the interaction of SPX domains with the coiled-coil motif of PHR transcription factors to regulate plant phosphate homeostasis

Martina K. Ried[1,7,9], Rebekka Wild [1,8,9], Jinsheng Zhu[1], Joka Pipercevic[2], Kristina Sturm[1], Larissa Broger[1], Robert K. Harmel[3,4], Luciano A. Abriata [5], Ludwig A. Hothorn[6,10], Dorothea Fiedler[3,4], Sebastian Hiller [2] & Michael Hothorn [1✉]

Phosphorus is an essential nutrient taken up by organisms in the form of inorganic phosphate (Pi). Eukaryotes have evolved sophisticated Pi sensing and signaling cascades, enabling them to stably maintain cellular Pi concentrations. Pi homeostasis is regulated by inositol pyrophosphate signaling molecules (PP-InsPs), which are sensed by SPX domain-containing proteins. In plants, PP-InsP-bound SPX receptors inactivate Myb coiled-coil (MYB-CC) Pi starvation response transcription factors (PHRs) by an unknown mechanism. Here we report that a $InsP_8$–SPX complex targets the plant-unique CC domain of PHRs. Crystal structures of the CC domain reveal an unusual four-stranded anti-parallel arrangement. Interface mutations in the CC domain yield monomeric PHR1, which is no longer able to bind DNA with high affinity. Mutation of conserved basic residues located at the surface of the CC domain disrupt interaction with the SPX receptor in vitro and in planta, resulting in constitutive Pi starvation responses. Together, our findings suggest that $InsP_8$ regulates plant Pi homeostasis by controlling the oligomeric state and hence the promoter binding capability of PHRs via their SPX receptors.

[1] Structural Plant Biology Laboratory, Department of Botany and Plant Biology, University of Geneva, 1211 Geneva, Switzerland. [2] Biozentrum Basel, 4056 Basel, Switzerland. [3] Leibniz-Forschungsinstitut für Molekulare Pharmakologie, 13125 Berlin, Germany. [4] Department of Chemistry, Humboldt-Universität zu Berlin, 12489 Berlin, Germany. [5] Protein production and structure Core Facility, EPFL, 1015 Lausanne, Switzerland. [6] Institute of Biostatistics, Leibniz University, 30419 Hannover, Germany. [7] Present address: Leibniz Institute of Plant Biochemistry, 06120 Halle, Germany. [8] Present address: Institut de Biologie Structurale (IBS), 38044 Grenoble, France. [9] These authors contributed equally: Martina K. Ried, Rebekka Wild. [10] Ludwig A. Hothorn is retired.
✉email: michael.hothorn@unige.ch

Phosphorus is an essential building block for many cellular components such as nucleic acids and membranes. It is essential for energy transfer and storage, and can act as a signaling molecule. Pro- and eukaryotes have evolved intricate systems to acquire phosphorus in the form of inorganic phosphate (Pi), to maintain cytosolic Pi concentrations and to transport and store Pi as needed. In green algae and plants, transcription factors have been previously identified as master regulators of Pi homeostasis and Pi starvation responses (PSR)[1,2]. Phosphorus starvation response 1 (CrPsr1) from *Chlamydomonas* and PHOSPHATE STARVATION RESPONSE 1 (AtPHR1) from *Arabidopsis* were founding members of plant-unique MYB-type coiled-coil (MYB-CC) transcription factors[3]. PHR transcription factors were subsequently characterized as regulators of PSRs in diverse plant species[4–6]. In *Arabidopsis*, there are 15 MYB-CCs with PHR1 and PHL1 controlling the majority of the transcriptional PSRs[7]. Knockout mutations in *Arabidopsis thaliana PHR1* (*AtPHR1*) result in impaired responsiveness of Pi starvation induced (*PSI*) genes and perturbed anthocyanin accumulation, carbohydrate metabolism, and lipid composition[2,8,9]. Overexpression of *AtPHR1* causes elevated cellular Pi concentrations and impacts the transcript levels of At*PHO2*, which codes for an E2 ubiquitin conjugase involved in PSR, via increased production of its micro RNA miR399d[8,10]. PHR binds to a GNATATNC motif (P1BS), found highly enriched in the promoters of PSI genes and in other *cis*-regulatory motifs, activating gene expression[2,7]. AtPHR1 is not only implicated in Pi homeostasis but also in sulfate, iron, and zinc homeostasis, as well as in the adaption to high-light stress[11–14]. Moreover, AtPHR1 shapes the plant root microbiome by negatively regulating plant immunity[15].

AtPHR1 and OsPHR2 have been previously reported to physically interact with stand-alone SPX proteins[16–19], additional components of PSR in plants[17,20–22]. SPX proteins may regulate PHR function by binding to PHRs under Pi-sufficient condition, keeping the transcription factor from entering the nucleus[23–25]. Alternatively, binding of SPX proteins to PHRs may reduce the ability of the transcription factors to interact with their promoter core sequences[18,19,23,24,26]. Two mechanisms were put forward regarding the regulation of the SPX–PHR interaction in response to changes in nutrient availability: SPX domains were proposed to act as direct Pi sensors, with the SPX–PHR interaction occurring in the presence of millimolar concentrations of Pi[18,19]. Alternatively, the integrity of the SPX–PHR complex could be regulated by protein degradation. Indeed, SPX degradation via the 26S proteasome is increased under Pi starvation[23,24,27].

Fungal, plant, and human SPX domains[28] have been independently characterized as cellular receptors for inositol pyrophosphates (PP-InsPs), which bind SPX domains with high affinity and selectivity[29,30]. PP-InsPs consist of a fully phosphorylated *myo*-inositol ring, carrying one or two pyrophosphate groups at the C1 and/or C5 position, respectively[31]. In plants, inositol 1,3,4-trisphosphate 5/6-kinase catalyzes the phosphorylation of phytic acid ($InsP_6$) to $5PP-InsP_5$ ($InsP_7$ hereafter)[32]. The diphosphoinositol pentakisphosphate kinases VIH1 and VIH2 then generate $1,5(PP)_2-InsP_4$ ($InsP_8$ hereafter) from $InsP_7$[29,33–35]. Plant diphosphoinositol pentakisphosphate kinases have been genetically characterized to play a role in jasmonate perception and plant defense responses[34] and, importantly, in nutrient sensing in *Chlamydomonas*[36] and *Arabidopsis*[29,35]. *vih1 vih2* double mutants lack the PP-InsP messenger $InsP_8$, over accumulate Pi, and show constitutive PSI gene expression[29,35]. A *vih1 vih2 phr1 phl1* quadruple mutant rescues the *vih1 vih2* seedling phenotypes and displays wild-type-like Pi levels, suggesting that VIH1, VIH2, PP-InsPs, and PHRs are part of a common signaling pathway[35]. In line with, the AtSPX1–AtPHR1 interaction is reduced in *vih1 vih2* mutant plants when compared

to wild type[29]. Thus, biochemical and genetic evidence implicates $InsP_8$ in the formation of a SPX–PHR complex[29,35].

Cellular $InsP_8$ pools are regulated by nutrient availability at the level of the VIH enzymes themselves. Plant VIH1 and VIH2, and diphosphoinositol pentakisphosphate kinases from other organisms are bifunctional enzymes, with an N-terminal kinase domain that generates $InsP_8$ from $InsP_7$ and a C-terminal phosphatase domain that hydrolyzes $InsP_8$ to $InsP_7$ and $InsP_6$[35,37,38]. The relative enzymatic activities of the two domains are regulated in the context of the full-length enzyme: under Pi starvation, cellular ATP levels are reduced, leading to a reduction of the VIH kinase activity and a reduction of $InsP_8$[29,35]. Pi itself acts as an allosteric regulator of the phosphatase activity[35,37]. Thus, under Pi-sufficient growth conditions, $InsP_8$ accumulates and triggers the formation of a SPX–$InsP_8$–PHR complex. Under Pi starvation, $InsP_8$ levels drop and the complex dissociates[29].

How the $InsP_8$-bound SPX receptor inactivates PHR function remains to be understood at the mechanistic level. It has been previously reported that AtPHR1 binds P1BS as a dimer[2]. Addition of SPX domains reduces the DNA-binding capacity of PHRs as concluded from electrophoretic mobility shift assays (EMSAs)[18,19,23]. Qi et al.[26] reported that AtPHR1 recombinantly expressed as a maltose-binding protein (MBP) fusion protein forms monomers in solution and binds DNA. This process can be inhibited by preincubating the recombinant transcription factor with AtSPX1 in the presence of high concentrations of $InsP_6$[26]. A recent crystal structure of the AtPHR1 MYB domain in complex with a promoter core fragment supports a dimeric binding mode of MYB-CC transcription factors[39]. Here we investigate the oligomeric state of PHRs, their DNA-binding kinetics, and the targeting mechanism of the interacting SPX receptors.

## Results

### PP-InsPs trigger AtSPX1–AtPHR1 complex formation in yeast.
The interaction of AtSPX1 with AtPHR1 has been previously characterized in yeast two-hybrid assays[19]. We reproduced the interaction of full-length AtPHR1 and AtSPX1 (Fig. 1a), and verified that all four stand-alone AtSPX proteins (AtSPX1–4) interact with a AtPHR1 fragment ($AtPHR1^{226–360}$) that contains the MYB domain and the CC domain in yeast (Supplementary Fig. 1a). This is in line with previous findings, reporting interaction of SPX domains with larger PHR fragments also containing the MYB and CC domains ($AtSPX1–AtPHR1^{208–362}$ and $OsSPX1/2–OsPHR2^{231–426}$)[18,19].

We next tested whether the SPX–PHR interactions observed in yeast are mediated by endogenous PP-InsPs. The putative PP-InsP-binding surface in AtSPX1 was mapped by homology modeling, using the previously determined *Chaetomium thermophilum* Gde1–$InsP_6$ complex structure (PDB-ID 5IJJ) as template[30]. We replaced putative PP-InsP-binding residues from the previously identified phosphate binding cluster (PBC: $AtSPX1^{Y25, K29, K139}$) and Lysine (K) surface cluster (KSC: $AtSPX1^{K136, K140, K143}$)[30] with alanines (Supplementary Fig. 1b). The resulting $AtSPX1^{PBC}$ and $AtSPX1^{KSC}$ mutant proteins failed to interact with $AtPHR1^{226–360}$ in yeast two-hybrid assays, whereas mutation of a conserved lysine residue outside the putative PP-InsP-binding site (structural control, $AtSPX1^{K81}$) to alanine had no effect (Supplementary Fig. 1b). We next deleted the known yeast PP-InsP kinase Vip1, which converts $InsP_6$ to $1PP-InsP_5$ and $InsP_7$ to $InsP_8$, or the PP-InsP kinase Kcs1, which converts $InsP_6$ to $InsP_7$ and $1PP-InsP_5$ to $InsP_8$[40,41] (Supplementary Fig. 1c, d). We found that deletion of either kinase reduced the interaction between wild-type AtSPX1 and $AtPHR1^{226–360}$ (Supplementary Fig. 1c). The interaction between the plant brassinosteroid receptor kinase BRI1 and the inhibitor protein BKI1, known to occur independently of

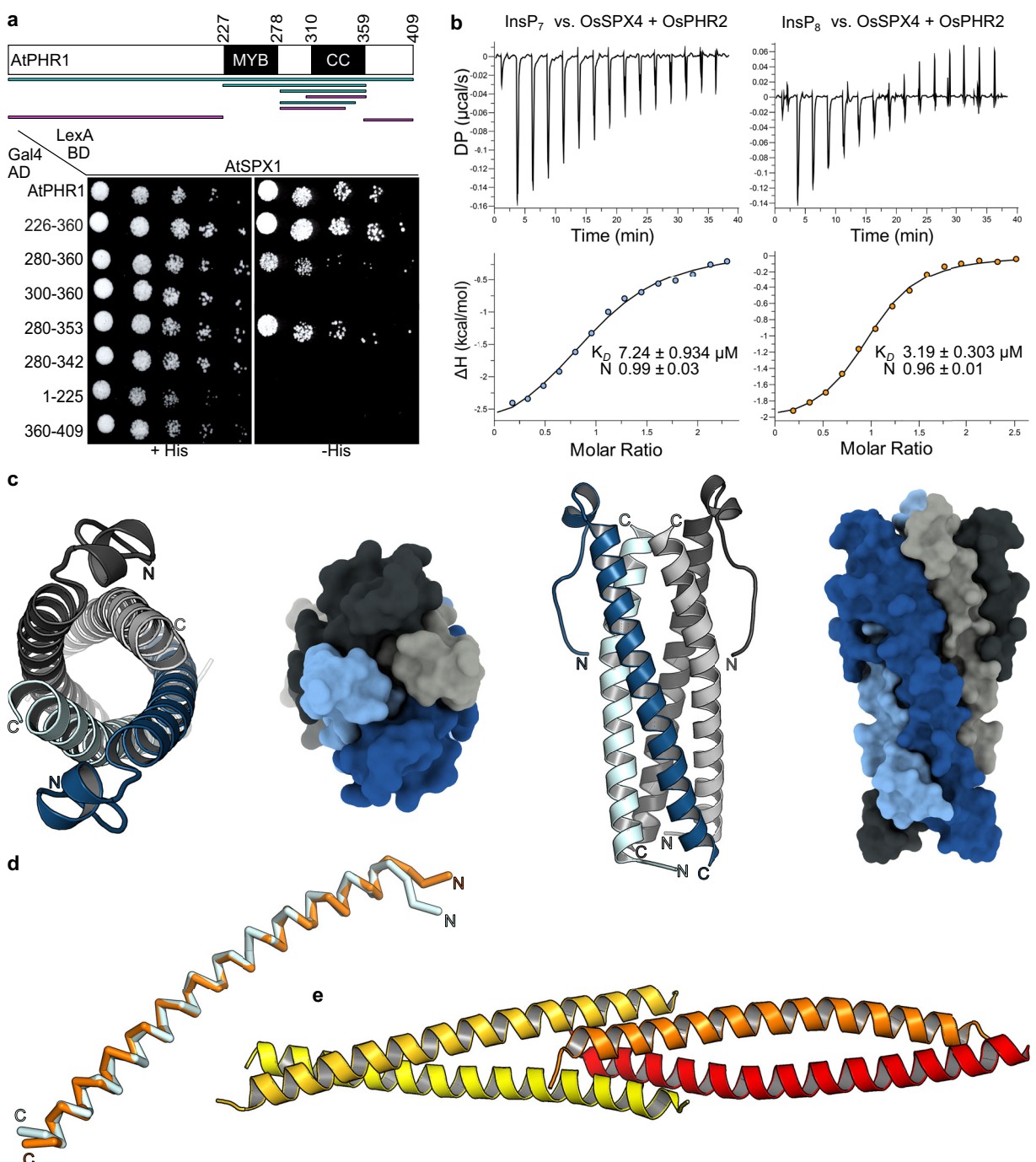

**Fig. 1 AtSPX1 recognizes the AtPHR1 coiled-coil domain that crystallizes as a tetramer. a** Yeast co-expressing different AtPHR1 deletion constructs fused to the Gal4-activation domain (AD; prey) and full-length wild-type AtSPX1 fused to the LexA-binding domain (BD; bait) were grown on selective SD medium supplemented with histidine (+His; co-transformation control) or lacking histidine (−His; interaction assay) to map a minimal fragment of AtPHR1 sufficient for interaction with AtSPX1. Shown are serial dilutions from left to right. A schematic overview of the tested interacting (in cyan) and non-interacting (in magenta) AtPHR1 fragments is shown alongside (MYB, MYB–DNA-binding domain; CC, coiled-coil domain). **b** Isothermal titration calorimetry assays of InsP$_7$ (400 μM 5PP-InsP$_5$; left panel) and InsP$_8$ (500 μM 1,5(PP)$_2$-InsP$_4$; right panel) binding to OsSPX4–OsPHR2 (30 μM), respectively. Raw heats per injection are shown in the top panel and the bottom panel represents the integrated heats of each injection, fitted to a one-site binding model (solid line). The insets show the dissociation constant ($K_D$) and binding stoichiometry (N) (±fitting error). **c** Ribbon and surface diagrams of the AtPHR1 CC four-stranded anti-parallel tetramer. Helices contributing to the dimer interface are shown in light and dark blue, respectively. Corresponding, symmetry-related helices completing the tetramer are shown in light and dark gray. **d** Structural superposition of two AtPHR1 core CC helices ($C_\alpha$ trace, in light blue) and ScCtp1 (PDB-ID 4×01, in orange)[44]. R.m.s.d. is ~1 Å comparing 45 corresponding $C_\alpha$ atoms. **e** Ribbon diagram of the ScCtp1 dimer-of-dimers CC domain, with contributing helices colored from yellow to red.

PP-InsPs[42], was not affected in either Δvip1 or Δkcs1 mutants (Supplementary Fig. 1c).

Using quantitative isothermal titration calorimetry (ITC)-binding assays, we have previously determined dissociation constants ($K_D$) for InsP$_6$ and InsP$_7$ binding to a OsSPX4–OsPHR2 complex to be ~50 and ~7 μM, respectively[30]. A side-by-side comparison of InsP$_7$ and InsP$_8$ binding to OsSPX4–OsPHR2 by ITC revealed dissociation constants of ~7 and ~3 μM, respectively (Fig. 1b). Taken together, the SPX–PHR interaction is mediated by PP-InsPs, with the bona fide Pi signaling molecule InsP$_8$ being the preferred ligand in vitro.

**AtSPX1 interacts with a unique four-stranded coiled-coil domain in AtPHR1.** We next mapped the SPX–PP-InsP-binding site in AtPHR1 to a fragment (AtPHR1$^{280-353}$), which comprises the CC domain and a 30-amino acid spanning N-terminal extension, in yeast two-hybrid experiments (Fig. 1a). We sought to crystallize an AtSPX1–PP-InsP–AtPHR1 complex either in the pre- or absence of P1BS fragments. We obtained crystals of a putative AtSPX1–InsP$_8$–AtPHR1$^{280-360}$ complex diffracting to 2.4 Å resolution and solved the structure by molecular replacement, using isolated SPX domain structures as search models[30]. Iterative cycles of model building and crystallographic refinement yielded, to our surprise, a well-refined model of AtPHR1$^{280-360}$ only (see "Methods"). Analysis with the program PISA revealed the presence of a crystallographic tetramer in which four long α-helices fold into an unusual anti-parallel four-stranded coiled-coil (Fig. 1c). AtPHR1$^{280-360}$ residues 292–356 and 310–357 are visible in the electron density maps from chain A and B, respectively. Residues 292–311 in chain A fold into a protruding loop region that harbors a small α-helix, and appear disordered in chain B (Fig. 1c). The anti-parallel α-helices in AtPHR1 closely align with a root mean square deviation (r.m.s.d.) of ~0.5 Å comparing 45 corresponding C$_α$ atoms. Structural homology searches with the program DALI[43] returned different coiled-coil structures, with a monomer of the tetrameric coiled-coil domain of the yeast transcription factor Ctp1 representing the closest hit (DALI Z-score 5.9, r.m.s.d. is ~1 Å comparing 45 corresponding C$_α$ atoms) (Fig. 1d)[44]. However, no anti-parallel four-stranded coiled-coil domain with structural similarity to AtPHR1 was recovered, with, for example, the Ctp1 dimer-of-dimers domain having a very different configuration (Fig. 1e)[44].

We next assessed the oligomeric state of AtPHR$^{280-360}$ using size-exclusion chromatography coupled to right-angle light scattering (SEC-RALS) and determined an apparent molecular weight of ~37.5 kDa, thus confirming that the isolated AtPHR1 CC forms tetramers in solution (theoretical molecular weight of the monomer is ~9.5 kDa) (Fig. 2a). Two additional crystal structures of AtPHR1$^{280-360}$ obtained in different crystal lattices all revealed highly similar tetrameric arrangements (Supplementary Fig. 2 and Supplementary Table 5). Although we found purified full-length AtPHR1 to be too unstable for SEC-RALS analysis, full-length OsPHR2 formed a tetramer in solution (Supplementary Fig. 3). In contrast, untagged AtPHR1$^{222-358}$, which comprises the CC and the MYB domains only, runs as a dimer (Fig. 2a, b black traces), in agreement with earlier reports[2].

**Mutations in the CC domain abolish AtPHR1 oligomerization and DNA binding in vitro.** The observed oligomeric state differences between full-length OsPHR2, AtPHR1$^{280-360}$ (CC), and AtPHR1$^{222-358}$ (MYB-CC) prompted us to investigate the putative dimer- and tetramerization interfaces in our AtPHR1 structures with the program PISA[45]. We found the dimerization (~1400 Å$^2$ buried surface area) and the tetramerization (~1900 Å$^2$ buried surface area) interfaces to be mainly

formed by hydrophobic interactions (Supplementary Fig. 4a, b). Both interfaces are further stabilized by hydrogen bond interactions and several salt bridges (Supplementary Fig. 4a, b). Importantly, all contributing amino-acids represent sequence fingerprints of the plant-unique MYB-CC transcription factor subfamily and are highly conserved among different plant species (Supplementary Fig. 4c). We identified residues specifically contributing to the formation of a CC dimer (Olig1: AtPHR1$^{L319}$, AtPHR1$^{I333}$, AtPHR1$^{L337}$, shown in cyan in Fig. 2 and Supplementary Fig. 4) or tetramer (Olig2: AtPHR1$^{L317}$, AtPHR1$^{L327}$, AtPHR1$^{I341}$, shown in dark orange in Fig. 2 and Supplementary Fig. 4) in our different CC structures (Supplementary Table 5). We replaced these residues by asparagine to generate two triple mutants in AtPHR1$^{222-358}$ and AtPHR1$^{280-360}$, respectively. We found in SEC-RALS assays that both mutant combinations dissolved AtPHR1$^{280-360}$ tetramers and AtPHR1$^{222-358}$ dimers into monomers, respectively (Fig. 2a, b).

Analysis of the secondary structure content of wild-type AtPHR$^{280-360}$ using circular dichroism (CD) spectroscopy revealed a 100% α-helical protein (Supplementary Fig. 5a), in agreement with our structural model of the AtPHR1 CC domain (Fig. 1c). In contrast, we estimated the secondary structure content of the Olig1 and Olig2 mutant proteins to be ~50% α-helical and ~50 random coil (Supplementary Fig. 5a). The CD melting spectrum for wild-type AtPHR$^{280-360}$ indicated the presence of a well-folded protein with a melting temperature ($T_m$) of ~50 °C, while we could not reliably determine $T_m$'s for the Olig1 and Olig2 mutant proteins (Supplementary Fig. 5b). We conclude that mutation of either AtPHR1$^{L319}$, AtPHR1$^{I333}$, AtPHR1$^{L337}$ or AtPHR1$^{L317}$, AtPHR1$^{L327}$, AtPHR1$^{I341}$ to asparagine disrupts the tetrameric coiled-coil domain of AtPHR1 and affects the structural integrity of the contributing α-helix.

It has been recently reported that the AtPHR1 MYB domain associates with its target DNA as a dimer[39]. We thus studied the capacity of AtPHR1$^{222-358}$ oligomerization mutants to interact with the P1BS in qualitative EMSA and quantitative grating-coupled interferometry (GCI) assays. AtPHR1$^{222-358}$ $^{Olig1}$ could still interact with the P1BS in EMSAs indistinguishable from wild type (Fig. 2c). However, AtPHR1$^{222-358}$ $^{Olig1}$ and AtPHR1$^{222-358}$ $^{Olig2}$ bound a biotinylated P1BS immobilized on the GCI chip with ~20-fold reduced affinity when compared to the wild-type control (Fig. 2d–f). Together, our experiments suggest that PHR1 may exist as a tetramer or dimer in solution, and that disruption of its plant-unique CC domain interface reduces the capacity of the transcription factor to bind its DNA recognition site.

**CC surface mutations abolish PHR–SPX interactions but do not interfere with DNA binding in vitro.** We next sought to identify the binding site for SPX in the PHR CC domain. In our structures, a conserved set of basic residues maps to the surface of the four CC helices (shown in magenta in Fig. 3a and Supplementary Fig. 4c). A similar set of surface-exposed basic residues has been previously found to form the binding site for PP-InsPs in various SPX receptors[30]. Mutation of AtPHR1$^{K325}$, AtPHR1$^{H328}$, and AtPHR1$^{R335}$, but not of AtPHR1$^{K308}$, AtPHR1$^{R318}$, and AtPHR1$^{R340}$ to alanine, disrupted the interaction of AtPHR1 with AtSPX1 in yeast (Fig. 3b). We simultaneously mutated the residues corresponding to AtPHR1$^{K325}$, AtPHR1$^{H328}$, and AtPHR1$^{R335}$ to alanine in OsPHR2 (OsPHR2$^{KHR/A}$). The mutant transcription factor showed no detectable binding to OsSPX4-InsP$_7$ in quantitative ITC assays, but maintained the ability to bind the P1BS (Fig. 3c, d). In line with this, mutation of the KHR motif does not alter the oligomeric state of AtPHR1$^{280-360}$ as concluded from SEC-RALS experiments (Fig. 2a magenta traces).

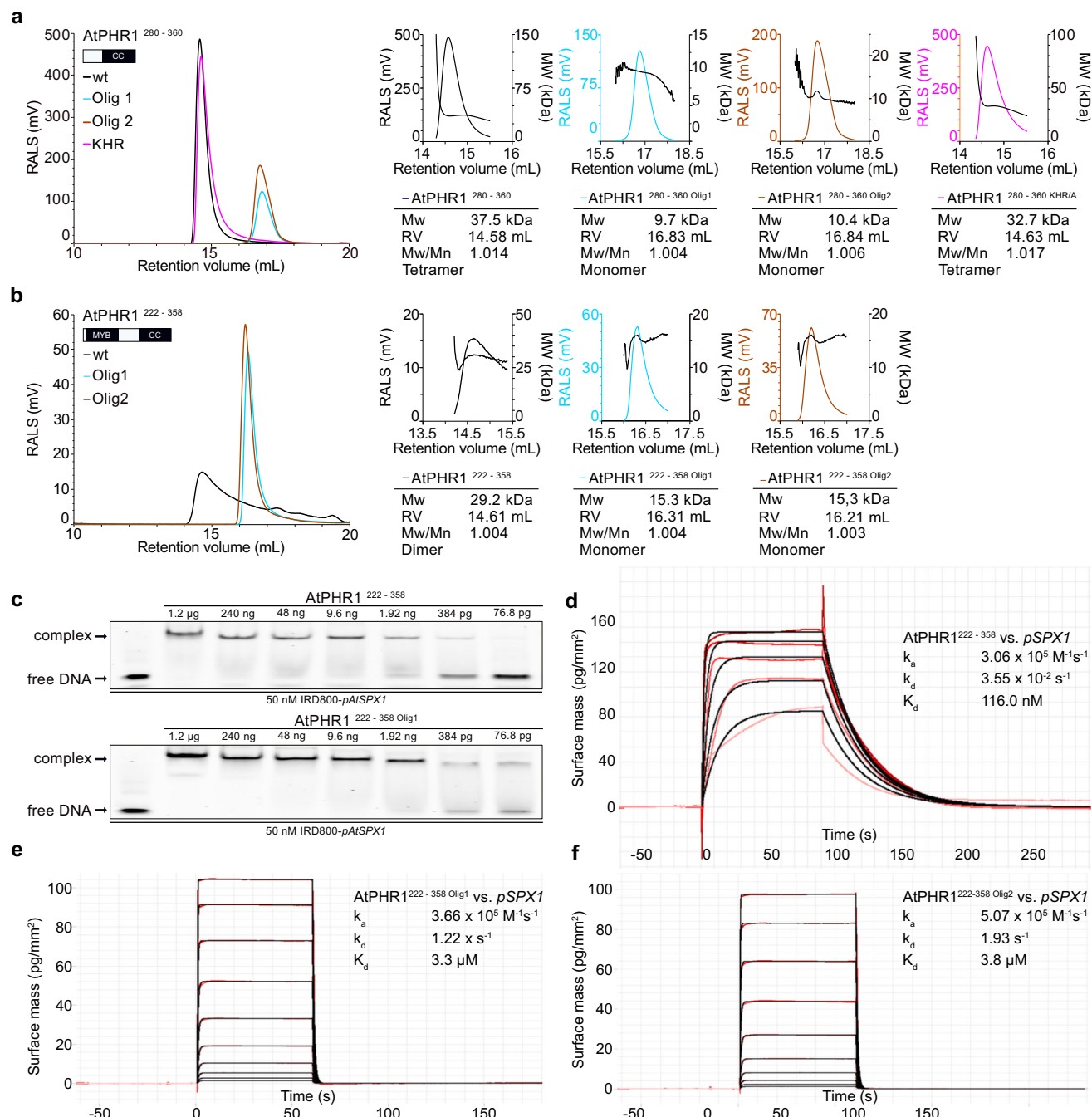

**Fig. 2 Mutations in the AtPHR1 coiled-coil domain impair oligomerisation and DNA binding. a** Analytical size-exclusion chromatography traces of wild-type AtPHR1 CC (wt, black line), AtPHR1[280–360] Olig1 (Olig1, cyan), AtPHR1[280–360] Olig2 (Olig2, orange), and of AtPHR1[280–360] KHR/A (KHR, magenta). The corresponding right-angle light scattering (RALS) traces are shown alongside, the molecular masses are depicted by a black line. Table summaries provide the molecular weight (Mw), retention volume (RV), dispersity (Mw/Mn), and the derived oligomeric state of the respective sample. **b** Analysis of AtPHR1 MYB-CC (AtPHR1[222-358]) as described in **a**. **c** Qualitative comparison of the interaction of AtPHR1[222-358] (upper panel) or AtPHR1[222-358] Olig1 (lower panel) binding to IRD800-*pAtSPX1* in electrophoretic mobility shift assays. Experiment was performed twice with similar results. **d–f** Quantitative comparison of the interaction of AtPHR1[222-358], AtPHR1[222-358 Olig1], or AtPHR12[2-358 Olig2] with *pSPX1* by grating-coupled interferometry (GCI). Sensorgrams show raw data (red lines) and their respective fits (black lines). Table summaries provide the derived association rate ($k_a$), the dissociation rate ($k_d$), and the dissociation constant ($K_d$).

Using nuclear magnetic resonance (NMR) spectroscopy, we next tested whether the KHR motif in AtPHR1 is directly involved in PP-InsP ligand recognition. We titrated InsP$_8$ into $^{15}$N, $^2$H-labeled AtPHR1[280–360], and recorded TROSY spectra using potassium phosphate (KPi) as control. Five backbone amide moieties exhibited chemical shift perturbations in the presence of InsP$_8$ but not in the presence of the KPi control. We acquired titration spectra using increasing concentrations of InsP$_8$ and estimated dissociation constants for InsP$_8$ based on three representative peaks (Supplementary Fig. 6). The derived dissociation constant is in the millimolar range and saturation could not be reached in the available concentration range

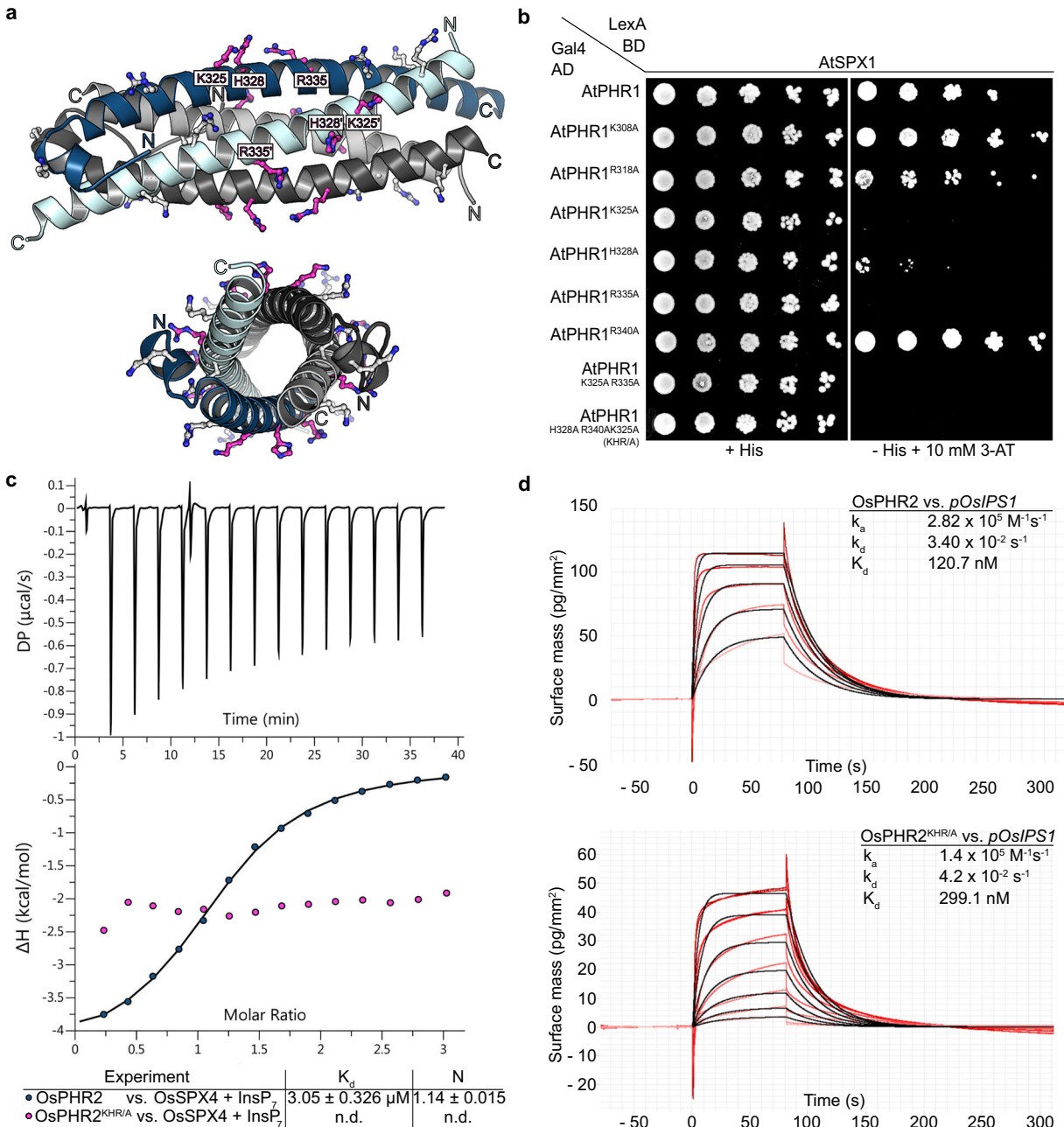

**Fig. 3 The KHR motif at the surface of the PHR coiled-coil domain is required for the interaction with SPX domains. a** Ribbon diagram of the AtPHR1 CC domain with conserved basic residues located at the surface of the domain shown in bonds representation. The KHR motif (AtPHR1[K325], AtPHR1[H328], AtPHR1[R335]) is highlighted in magenta. **b** Mutational analysis of the basic residues in AtPHR1 CC. Yeast co-expressing AtPHR1[226-360] variants in which surface-exposed basic residues have been replaced with alanine fused to the Gal4-AD (prey) and AtSPX1 fused to the LexA-BD (bait) were grown on selective SD medium supplemented with histidine (+His; co-transformation control) or lacking histidine and supplemented with 10 mM 3-amino-1,2,4-triazole (3-AT) (−His + 10 mM 3-AT; interaction assay) to identify residues required for interaction with AtSPX1 in yeast two-hybrid assays. Shown are serial dilutions from lift to right. **c** Isothermal titration calorimetry (ITC) assay of wild-type OsPHR2 and OsPHR2[KHR/A] (300 μM) vs. OsSPX4 (20 μM)–5PP-InsP5 (100 μM). Raw heats per injection are shown in the top panel, the bottom panel represents the integrated heats of each injection, fitted to a one-site binding model (solid line). The insets show the dissociation constant ($K_D$) and binding stoichiometry ($N$) (± fitting error, n.d. no detectable binding). **d** Quantitative comparison of the interaction of OsPHR2 (top panel) or OsPHR2KHR/A (bottom panel) with *pOsIPS1* by GCI. Sensorgrams show raw data (red lines) and their respective fits (black lines). The insets show summarize association rates ($k_a$), dissociation rates ($k_d$), and the dissociation constant ($K_d$) of the respective sample.

(Supplementary Fig. 6). We next repeated the same set of experiments using the AtPHR1[KHR/A] mutant protein, located the same peaks in the transverse relaxation-optimized spectroscopy (TROSY) spectra and estimated similar dissociation constants (Supplementary Fig. 6).

Taken together, three highly conserved basic residues located at the surface of the PHR coiled-coil domain are critical for the interaction with the PP-InsP-bound SPX receptor (Supplementary Fig. 4c). The NMR titration experiments suggest that AtPHR1 does not contribute to the specific binding of InsP8 and

that the low affinity interaction between the CC domain and $InsP_8$ does not involve the KHR motif (Supplementary Fig. 6).

**Mutation of the AtPHR1 KHR motif impairs AtSPX1 binding and Pi homeostasis in planta.** We next tested whether mutation of the SPX-binding site in AtPHR1 can modulate its function in Pi homeostasis in *Arabidopsis*. We expressed wild-type and point-mutant versions of AtPHR1 carrying an N-terminal FLAG tag under the control of its native promoter in a *phr1-3* loss-of-function mutant[8]. At seedling stage, we found that AtPHR1 single, double, and triple point mutations complemented the previously characterized Pi deficiency phenotype of *phr1-3*[2,8]. (Fig. 4a and Supplementary Fig. 7a). After transferring the seedlings to soil, variable growth phenotypes became apparent 21 days after germination (DAG) (Supplementary Fig 7b). From three independent lines per genotype, we selected one line each showing similar *AtPHR1* transcript levels for all experiments shown in Fig. 4 (Fig. 4 and Supplementary Fig. 8a). Comparing these lines, we found that $AtPHR1^{K325A,R335A}$ double and $AtPHR1^{K325A,H328A,R335A}$ triple mutants, but not the single mutants, displayed severe growth phenotypes, with the triple mutant showing the strongest defects (Fig. 4a). We next determined cellular Pi levels in all independent lines and found that (i) Pi levels are positively correlated with AtPHR1 expression levels (Supplementary Fig. 8a), that (ii) all AtPHR1 mutant proteins tested accumulate Pi to significantly higher levels when compared to wild type and *phr1-3*, and that (iii) the $AtPHR1^{K325A,H328A,R335A}$ triple mutant displayed the highest Pi levels (Fig. 4b and Supplementary Fig. 8b–d). In line with this, PSI gene expression is misregulated in $AtPHR1^{K325A,R335A}$ double and $AtPHR1^{K325A,H328A,R335A}$ ($AtPHR1^{KHR/A}$) triple mutants (Fig. 4c). In co-immunoprecipitation assays in *Nicotiana benthamiana* and in *Arabidopsis*, we found the interaction of $AtPHR1^{K325A,H328A,R335A}$ with AtSPX1 to be reduced when compared to wild-type AtPHR1 (Fig. 4d and Supplementary Fig. 8).

We next studied the genetic interaction between *PHR1* and *VIH1/2*. As previously reported, the severe phenotypes of *vih1-2 vih2-4* seedlings are partially rescued in the *phr1 phl1 vih1-2 vih2-4* quadruple mutant, suggesting that VIH1/2-generated $InsP_8$ regulates the activity of PHR1 and PHL1 by promoting the binding of SPX receptors[29,35]. We performed the orthogonal genetic experiment, by complementing the *phr1 phl1* mutant with $AtPHR1^{KHR/A}$ expressed under the control of the AtPHR1 promoter and carrying a N-terminal enhanced green fluorescent protein (eGFP) tag (Fig. 4e, see "Methods"). The complemented lines displayed intermediate growth phenotypes and constitutive PSI gene expression (Fig. 4e, f). Thus, SPX–$InsP_8$ mediated regulation of PHR1 and PHL1 has to be considered one of several PP-InsP regulated processes affected in the *vih1-2 vih2-4* mutant. Together, our in vivo experiments reveal that SPX receptors interact with the CC domain of AtPHR1 via the surface-exposed Lys325, His328, and Arg335, and that this interaction negatively regulates PHR activity and PSRs.

## Discussion
PHR transcription factors have been early on recognized as central components of the PSR in green algae and in plants, directly regulating the expression of PSI genes[1,2,7]. In *Arabidopsis* and in rice *spx* mutants of then unknown function also showed altered PSI gene expression[20,22]. This genetic interaction was later substantiated by demonstrating that stand-alone plant SPX proteins can interact with PHR orthologs from *Arabidopsis* and rice[18,19,23]. The biochemical characterization of SPX domains as cellular receptors for PP-InsPs and the genetic identification of VIH kinases

as master regulators of PSR in plants suggested that PP-InsPs, and specifically $InsP_8$ mediates the interaction of SPX proteins with PHRs in response to changing nutrient conditions[29,30,35].

Our quantitative DNA-binding assays demonstrate that AtPHR1 MYB-CC binds the P1BS from the *AtSPX1* promoter with high affinity ($K_d \sim 0.2\,\mu M$), in agreement with previously reported binding constants for different MYB-CC constructs ($K_d \sim 0.01$–$0.1\,\mu M$)[26,39]. Different oligomeric states have been reported for various PHR MYB-CC constructs[2,26]. Our AtPHR1 MYB-CC construct behaves as a dimer in solution, consistent with the recently reported crystal structure of the AtPHR1 MYB–DNA complex and with earlier reports[2,39]. Purified full-length OsPHR2, however, appears to be a homotetramer in solution (Supplementary Fig. 3).

In yeast two-hybrid assays, we found that AtSPX1–4 all are able to interact with AtPHR1 (Fig. 1a and Supplementary Fig. 1a). We mapped their conserved interaction surface to the plant-unique CC motif of PHRs (Fig. 1a). Crystal structures of this fragment reveal an unusual, four-stranded anti-parallel coiled-coil domain (Fig. 1c). Given the fact, that AtPHR1 MYB-CC forms dimers in solution (Fig. 2b), we cannot exclude the possibility that the CC tetramers represent crystal packing artifacts. However, we did observe identical CC tetramers in three independent crystal lattices (Supplementary Fig. 2) and in solution (Fig. 2a). The residues contributing to the dimer and to the tetramer interfaces are highly conserved among all plant MYB-CC transcription factors (Supplementary Fig. 4). Mutation of either interface blocks AtPHR1 oligomerization in vitro (Fig. 2a, b) and reduces DNA binding (Fig. 2d–f). An attractive hypothesis would thus be that AtPHR1 binds its target promoter as a dimer, but can potentially form homo-tetramers, or hetero-tetramers with other MYB-CC type transcription factors sharing the conserved, plant-unique CC structure and sequence (Supplementary Fig. 4). Notably, PHR1 PHL1 heteromers have been previously described[7].

We found that SPX–PHR complex formation is mediated by endogenous PP-InsPs in yeast cells, as deletion of the yeast PP-InsP kinases Vip1 and Kcs1 abolished the interaction, and mutation of the PP-InsP-binding surface in AtSPX1 interfered with AtPHR1 binding (Supplementary Fig. 1b, c). In line with this, SPX–PHR complexes are found dissociated in *vih1 vih2* mutant plants[29]. It is of note that the observed differences in binding affinity for $InsP_7$ and $InsP_8$ to SPX–PHR in vitro (Fig. 1b)[29] cannot fully rationalize the apparent preference for $InsP_8$ in vivo[29,35]. We identified the binding surface for SPX–$InsP_8$ by locating a set of highly conserved basic residues exposed at the surface of the CC domain (Fig. 3a and Supplementary Fig. 4). Mutation of this KHR motif did not strongly impact the ability of isolated OsPHR2 to bind *pOsIPS1* in vitro (Fig. 3d), but disrupted the interaction of AtPHR1 with AtSPX1 in yeast (Fig. 3b). The corresponding mutations in OsPHR2 had a similar effect on the interaction with OsSPX4 in quantitative ITC assays (Fig. 3c). Expression of $AtPHR1^{KHR/A}$ in the *phr1-3* mutant resulted in Pi hyperaccumlation phenotypes and constitutive PSI gene expression in Arabidopsis (Fig. 4a–c). The intermediate growth phenotypes of *vih1 vih2 phr1 phl1* mutants complemented with $AtPHR1^{KHR/A}$ clearly suggests that PP-InsPs do not only regulate the activity of PHR1 and PHL1 in plants, but likely the function of other (SPX domain-containing) proteins[30] (Fig. 4e). Notably, binding of $AtPHR1^{KHR/A}$ to AtSPX1 was reduced in co-immunoprecipitation assays when compared to wild-type AtSPX1 (Fig. 4d and Supplementary Fig. 9). Thus, our and previous finding suggest that $InsP_8$ can promote the association of SPX receptors and PHR transcription factors. The newly identified basic surface area in PHR CC, harboring the conserved KHR motif, likely forms part of the SPX–PHR complex interface (Fig. 3a).

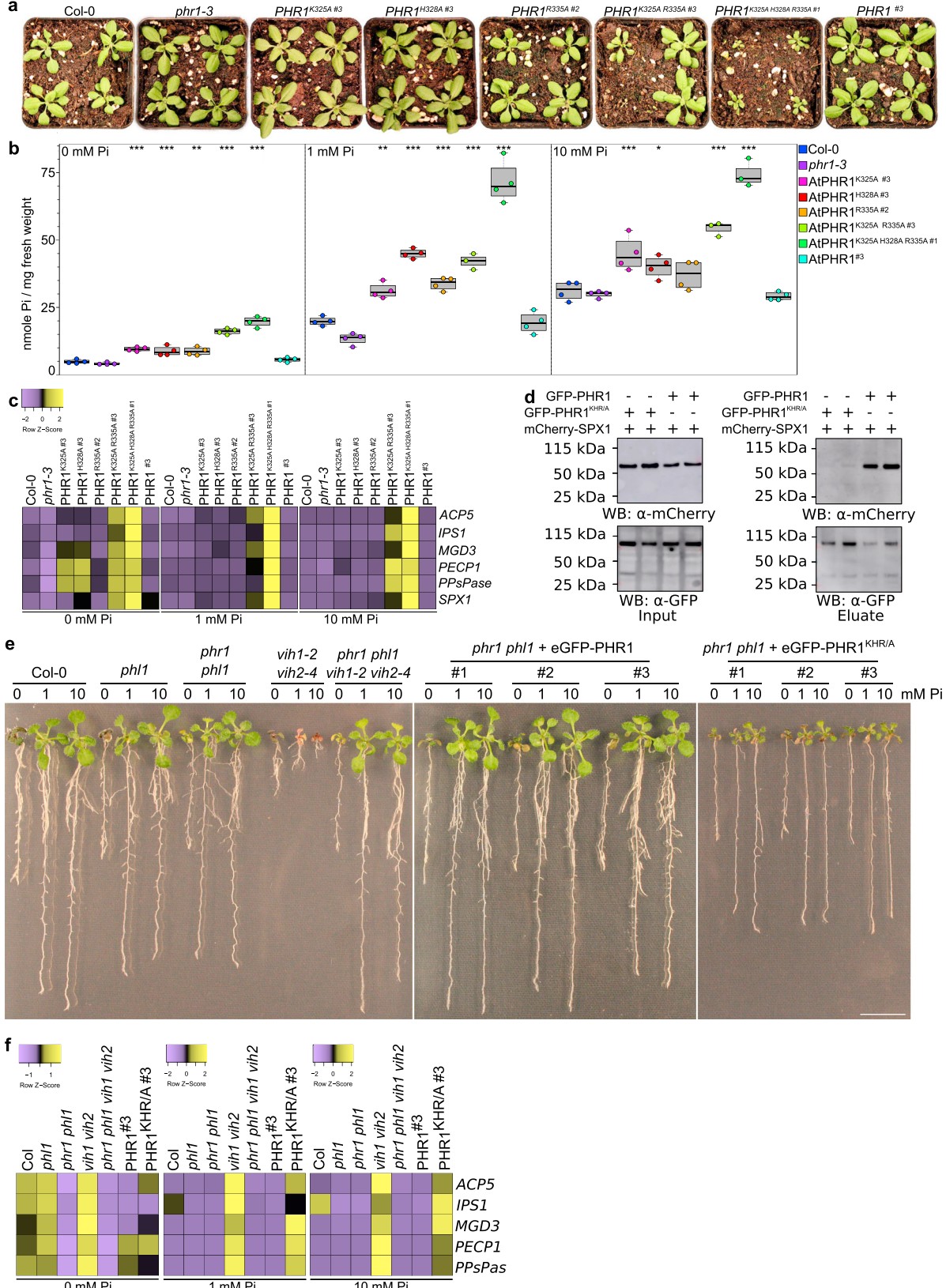

It is of note that addition of AtSPX1 to AtPHR1 has been previously demonstrated to reduce AtPHR1's ability to bind to P1BS in the presence of InsP$_6$[26]. We could not quantify these interactions in ITC or GCI binding assays, as PHR CC formation is much preferred over SPX–InsP$_8$ binding at the protein concentrations required in these assays. We speculate that InsP$_8$ bound SPX proteins can bind to the basic residues we identified in the PHR CC domain to control the oligomeric state and hence the promoter binding capacity of PHRs. As these residues are conserved among all plant MYB-CC proteins this may suggest that transcription factors outside the PHR subfamily may be regulated by SPX domains and PP-InsPs, possibly rationalizing

**Fig. 4 Mutation of the AtPHR1 KHR motif impairs interaction with AtSPX1 and Pi homeostasis in planta. a** Growth phenotypes of Col-0 wild type, *phr1-3* of *phr1-3* complementation lines expressing FLAG-AtPHR1, FLAG-AtPHR1$^{K325A}$, FLAG-AtPHR1$^{H328A}$, FLAG-AtPHR1$^{R335A}$, FLAG-AtPHR1$^{K325A\,R335A}$, and FLAG-AtPHR1$^{K325A\,H328A\,R335A}$ under the control of the *AtPHR1* promoter at 21 DAG grown in Pi-sufficient conditions. One representative line per complementation construct is shown (specified by a #), additional lines are shown in Supplementary Fig. 4. **b** Plot representing the cellular Pi content of the lines shown in **a**. Bold black line, median; box, interquartile range (IQR); whiskers, lowest/highest data point within 1.5 IQR of the lower/upper quartile. Seedlings were germinated and grown on vertical ½MS plates for 8 days, transferred to ½MS plates supplemented with either 0, 1, or 10 mM Pi and grown for additional 7 days. For each line, four plants were measured in technical duplicates. Pi contents of all lines can be found in Supplementary Fig. 4 (*$p < 0.05$; **$p < 0.01$; ***$p < 0.001$). **c** Heat maps of PSI marker gene (*ACP5, IPS1, MGD3, PECP1, PPsPase, SPX1*) expression analyses of the lines shown in **a**, represented as *Z*-scores. For each line, three biological replicates were analyzed in technical triplicates by qRT-PCR. **d** Co-immunoprecipitation (Co-IP) experiment assessing the ability for immobilized GFP-AtPHR1 and GFP-AtPHR1$^{KHR/A}$ to interact with mCherry-AtSPX1 in *N. benthamiana*. Input western blottings are shown alongside. Experiment was performed twice with similar results. **e** Genetic interactions in the VIH-PHR signaling pathway. Col-0 wild type and the indicated mutant seedlings were grown on ½MS plates for 7 DAG, transferred to ½MS plates supplemented with either 0, 1, or 10 mM Pi and grown for additional 7 days. For complementation analyses, wild-type AtPHR1 or AtPHR1$^{KHR/A}$ was expressed as an N-terminal eGFP fusion protein under the control of the *AtPHR1* promoter. **f** Heat maps of PSI marker gene expression for the lines shown in **e**.

the severe phenotypes of *vih1 vih2* mutant plants (Fig. 4e and Supplementary Fig. 4). The recent findings that VIH kinases and PHRs act together in plant PSR and that SPX–PHR complexes are dissociated in *vih1 vih2* mutants further suggest that InsP$_8$ is the bona fide signaling molecule promoting the association between SPX domains and MYB-CCs[29,35]. Repressive SPX–PHR complexes consequently form only under Pi-sufficient conditions, where InsP$_8$ levels are high[29,35]. Under Pi starvation, when InsP$_8$ levels are reduced, SPX–PHR complexes dissociate, enabling the transcription factors to acquire the oligomeric state required for high affinity promoter binding. The physiological and mechanistic investigation of this central process may, in the long term, contribute to the development of Pi starvation resilient crops. This could in turn sustain the use of the essential and non-renewable resource rock phosphate, which is currently consumed at an alarming scale.

## Methods

**Molecular cloning, constructs, and primers**. For a detailed description of the cloning strategies, constructs and primers used in this study, please refer to Supplementary Tables 2 and 3.

**Generation of stable transgenic *A. thaliana* lines**. All stable transgenic *A. thaliana* lines are listed in Supplementary Table 1. Constructs were introduced into *Agrobacterium tumefaciens* strain pGV2260 and *A. thaliana* plants were transformed via floral dipping[46]. Transformants were identified by mCherry fluorescence with a Zeiss Axio Zoom.V16 stereo microscope (mRFP filter) and a HXP200C illuminator. Homozygous T3 lines have been identified for complementation lines expressing FLAG-AtPHR1, FLAG-AtPHR1$^{K325A}$, FLAG-AtPHR1$^{H328A}$, and FLAG-AtPHR1$^{R335A}$ under the control of the native *AtPHR1* promoter. For complementation lines expressing FLAG-AtPHR1$^{K325A\,R335A}$ and FLAG-AtPHR1$^{K325A\,H328A\,R335A}$ under the control of the native *AtPHR1* promoter, T2 lines were used throughout, and homozygous and heterozygous transformants were selected for all experiments by mCherry fluorescence as described above. T3 homozygous lines expressing eGFP-PHR1 under the control of the *AtPHR1* promoter were identified by their Hygromycin resistance. *PHR1* was amplified from *Arabidopsis* cDNA and introduced into pH7m34GW binary vector. Point mutations were introduced by site-directed mutagenesis (primers are listed in Supplementary Table 3).

**Yeast two-hybrid experiments**. (Screen) AtSPX1$^{1-252}$ was used as a bait and screened against an *A. thaliana* seedling cDNA library by Hybrigenics Services. AtSPX1$^{1-252}$ was cloned into the pB29 vector providing a C-terminal LexA-DNA-binding domain and transformed into yeast strain L40αGal4 (MATa). Prey genes were cloned into the pP6 vector providing a N-terminal Gal4-activation domain and transformed into yeast strain YHGX13 (MATα). After mating haploid bait and prey strains, positive interactions were detected by growth on histidine deficient medium.

*Yeast strains and media*. For all experiments, either the diploid TATA strain (Hybrigenics Services) or the haploid L40 strain was used (Supplementary Table 2). Cells were routinely maintained on yeast extract-peptone-adenine-dextrose (YPAD) plates (20 g/L glucose, 20 g/L bacto-peptone, 10 g/L yeast extract, 0.04 g/L adenine hemisulfate, and 20 g/L agar). Experiments were performed on synthetic dropout (SD) plates (6.7 g/L yeast nitrogen base with adenine hemisulfate

and without leucine, tryptophan, and histidine, 20 g/L glucose, and 20 g/L agar) supplemented with 0.076 g/L histidine or 10 mM 3-amino,1,2,4-triazole (3-AT).

*Yeast transformation*. One yeast colony was resuspended in 500 μL sterile H$_2$O, plated on YPAD plates, and grown for 2 days until the whole plate was covered with yeast. Yeast cells were then resuspended in 50 mL YPAD liquid medium and the OD$_{600nm}$ was determined ($2 \times 10^6$ cells/mL were used for one transformations). Cells were centrifuged at $3000 \times g$ and 4 °C for 5 min, resuspended in 25 mL Tris-EDTA (TE) buffer, centrifuged again at $3000 \times g$ and 4 °C for 5 min, resuspended in 2 mL LiAc/TE buffer, centrifuged at $16,000 \times g$ and room temperature (RT) for 15 s, and finally resuspended in 50 μL/transformation LiAc/TE buffer. The transformation mix (0.5 μg bait plasmid, 0.5 μg prey plasmid, 10 μL ssDNA (10 mg/mL), 50 μL yeast cells, 345 μL 40% (w/v) PEG3350 in LiAC/TE) was prepared and incubated at 30 °C for 45 min, followed by incubation at 42 °C for 30 min. Finally, yeast cells were centrifuged at $6500 \times g$ and RT for 15 s, resuspended in TE buffer, plated on SD plates lacking leucine and tryptophan, and incubated at 30 °C for 3 days.

*Yeast spotting dilution assay*. Positive transformants were selected on SD plates without tryptophan and leucine, and incubated at 30 °C for three days. Cells were counted, washed in sterile water and spotted in 5 times dilution (5000, 1000, 200, 40, and 8 cells) on SD plates without either tryptophan and leucine, or tryptophan, leucine, and histidine supplemented with 10 mM 3-AT. Plates were incubated at 30 °C for 3 days.

**Protein expression and purification**. For in vitro biochemistry, AtPHR1$^{280-360\,wt/KHR/Olig1/Olig2}$ and AtPHR1$^{222-358\,wt/Olig1/Olig2}$ were cloned into the pMH-HT protein expression vector, providing a N-terminal 6 × His affinity tag with a tobacco etch virus (TEV) protease recognition site. OsPHR2$^{1-426\,wt/KHR/A}$ was cloned into the pMH-HSgb1T protein expression vector, providing a N-terminal 8 × His-Strep-GB1 affinity tag with a TEV cleavage site. OsSPX1$^{1-321}$ was cloned into the pMH-HSsumo protein expression vector, providing a N-terminal 8xHis-Strep-Sumo affinity tag. All constructs were transformed into *Escherichia coli* BL21 (DE3) (*argU, ileY, leuW*) RIL cells. For recombinant protein expression, cells were grown at 37 °C in terrific broth (TB) medium to an OD$_{600nm}$ of ~0.6. After reducing the temperature to 18 °C, protein expression was induced with 0.3 mM isopropyl β-D-galactoside (IPTG) for 16 h. Cells were centrifuged at $4500 \times g$ and 4 °C for 1 h, resuspended in lysis buffer (50 mM Tris-HCl pH 7.8, 500 mM NaCl, 0.1% (v/v) IGEPAL, 1 mM MgCl, 2 mM β-mercaptoethanol), snap-frozen in liquid nitrogen, and stored at −80 °C. For protein preparation, cells were thawed, supplemented with cOmplete$^{TM}$ EDTA-free protease inhibitor cocktail (Roch), DNaseI, and lysozyme, and disrupted using a sonicator. Cell lysates were centrifuged at $7000 \times g$ and 4 °C for 1 h, sterile filtered, supplemented with 20 mM imidazole, and loaded onto a 5 mL HisTrap HP Ni$^{2+}$ affinity column (GE Healthcare). After washing with several column volumes (CVs) of lysis buffer supplemented with 20 mM imidazole, high-salt buffer (50 mM Tris-HCl pH 7.8, 1 M NaCl, 2 mM β-mercaptoethanol), and high-phosphate buffer (200 mM K$_2$HPO$_4$/KH$_2$PO$_4$ pH 7.8, 2 mM β-mercaptoethanol), proteins were eluted in a gradient from 20 to 500 mM imidazole in lysis buffer. The purified proteins were cleaved by TEV or Sumo protease overnight at 4 °C (1 : 100 ratio) and dialyzed against lysis buffer for PHR1$^{222-358\,wt/Olig1/Olig2}$ and PHR1$^{280-360\,wt/KHR/Olig1/Olig2}$ fragments, against modified lysis buffer (25 mM Tris-HCl pH 7.8, 300 mM NaCl, 0.1% (v/v) IGEPAL, 1 mM MgCl, 2 mM β-mercaptoethanol) for OsPHR2$^{1-426\,wt/KHR/A}$, and against modified anion exchange buffer (20 mM Tris-HCl pH 6.5, 500 mM NaCl) for OsSPX4$^{1-321}$. PHR1$^{280-360\,wt/KHR/Olig1/Olig2}$ and OsPHR2$^{1-426\,wt/KHR/A}$ were subjected to a second Ni$^{2+}$ affinity purification in either lysis buffer or modified lysis buffer, respectively, and the flow-throughs were collected and concentrated. PHR1$^{222-358\,wt/Olig1/Olig2}$ were subjected to cation exchange (50 mM HEPES pH 7.5, 50–1000 mM NaCl) and OsSPX4$^{1-321}$ was subjected to anion exchange (20 mM Tris-HCl, 50–1000 mM NaCl). Fractions corresponding to

the respective proteins were pooled and concentrated. All proteins were loaded onto a HiLoad Superdex 75 pg HR26/60 column (GE Healthcare), pre-equilibrated in gel filtration buffer A (20 mM Tris/HCl pH 7.5, 300 mM NaCl, 0.5 mM tris(2-carboxyethyl)phosphine (TCEP)) for PHR1$^{222-358\ wt/Olig1/Olig2}$, or in gel filtration buffer B (20 mM Tris/HCl pH 7, 200 mM NaCl, 0.5 mM TCEP) for the remaining proteins. Fractions containing the respective proteins were pooled and concentrated. Purified and concentrated protein was immediately used for further experiments or snap-frozen in liquid nitrogen and stored at −80 °C.

The AtPHR1$^{280-360}$ fragment used for crystallization was cloned into the pMH-HS-Sumo protein expression vector, providing a N-terminal 8 × His-StrepII tandem affinity tag and a Sumo fusion protein. The construct was transformed into E. coli BL21 (DE3) RIL cells. For recombinant protein expression, cells were grown at 37 °C in TB medium to an OD$_{600nm}$ of ~0.6. After reducing the temperature to 16 °C, protein expression was induced with 0.3 mM IPTG for 16 h. Cells were centrifuged for 20 min at 4000 × g and 4 °C, then the cell pellet was washed with PBS, snap-frozen in liquid nitrogen and stored at −80 °C. For the purification of a putative AtSPX1–InsP$_8$–AtPHR1$^{280-360}$ complex, the AtPHR1 cell pellet was thawed and mixed with twice the amount of cells expressing a His-Strep-MBP-AtSPX1$^{1-251}$ fusion protein, which provides a N-terminal, TEV-cleavable MBP. Lysis buffer (200 mM KP$_i$ pH 7.8, 2 mM β-mercaptoethanol) supplemented with 0.1% (v/v) IGEPAL, 1 mM MgCl$_2$, 10 mM imidazole, 500 units TurboNuclease (BioVision), 2 tablets Protease Inhibitor Cocktail (Roche), and 0.1 mM InsP$_8$ was added and cells were disrupted using an EmulsiFlex-C3 (Avestin). Cell lysates were centrifuged at 7000 × g and 4 °C for 1 h. The cleared supernatant was sterile filtered and loaded onto a 5 mL HisTrap HP Ni$^{2+}$ affinity column (GE Healthcare). After washing with several CVs of lysis buffer, the protein was eluted with 250 mM imidazole in lysis buffer. The His-Strep-Sumo-AtPHR1/His-Strep-MBP-AtSPX1 fusion proteins were cleaved by TEV and Sumo protease treatment overnight at 4 °C, while dialyzing in a buffer containing 200 mM KP$_i$ pH 7.8, 100 mM NaCl, 2 mM β-mercaptoethanol. Imidazole (10 mM) was added to the cleaved protein sample and a second Ni$^{2+}$ affinity step was performed in order to remove the cleaved-off His-Strep-Sumo/MBP fusion tags as well as the 6 × His-tagged Sumo and TEV proteases. The flow-through was concentrated and loaded onto a HiLoad Superdex 75 pg HR16/60 column (GE Healthcare), pre-equilibrated in gel filtration buffer (20 mM Tris/HCl pH 7.8, 250 mM NaCl, 2.5 mM InsP$_6$, 0.5 mM TCEP). Fractions containing the co-eluting AtSPX1$^{1-251}$ and AtPHR1$^{280-360}$ proteins were pooled and concentrated. A second SEC step was performed using a HiLoad Superdex 200 pg HR26/60 column (GE Healthcare) and the same gel filtration buffer as above. It was later found that AtSPX1$^{1-251}$ expressed in E. coli is largely unfolded and unable to bind InsP$_8$ and hence no complex with AtPHR1$^{280-360}$ was observed in our crystals. Purified and concentrated protein was immediately used for further experiments or snap-frozen in liquid nitrogen and stored at −80°C.

For production of [U-$^{15}$N,$^2$H]-labeled AtPHR1$^{280-360}$ variants for NMR spectroscopy, AtPHR1$^{280-360}$ and AtPHR1$^{280-360}$ $^{KHR/A}$ in pMH-HT were transformed and expressed in E. coli BL21 (DE3) RIL cells. Luria-Beltani liquid medium (2 mL) was inoculated with freshly transformed cells. After overnight growth at 37 °C, cells were transferred to M9$_{H2O}$ medium and grown overnight at 37 °C. From this culture, 150 μL were transferred into 1 mL M9$_{H2O}$ medium and grown at 37 °C overnight. Cells were transferred into 100 mL M9$_{D2O}$ medium and grown overnight at 37 °C. The culture was added to 0.9 L of M9$_{D2O}$ medium and grown at 37 °C to an OD$_{600nm}$ of ~0.4, then the temperature was shifted to 16 °C. After 1 h, protein expression was induced with 0.3 mM IPTG and cells were grown overnight, reaching a final OD$_{600nm}$ of 1.2–1.4. The cell culture was harvested at 7800 × g and the cell pellet was stored at −20 °C. The cell pellet was resuspended in 50 mM Tris pH 7.8, 500 mM NaCl, 0.1% IGEPAL, 1 mM MgCl$_2$, 2 mM β-mercaptoethanol, 20 mM imidazole, and DNAseI (AppliChem). The resuspension was homogenized by magnetic stirring and after 10 min lysozyme was added. The cells were lysed by sonication and subsequently centrifuged at 42,500 × g for 45 min. The collected supernatant was applied on 5 mL Histrap HP (GE Healthcare) equilibrated in a buffer A (50 mM Tris pH 7.8, 500 mM NaCl and 2 mM β-mercaptoethanol) and washed with a wash buffer 1 (50 mM Tris pH 7.8, 200 mM KPi, and 2 mM β-mercaptoethanol) and with wash buffer 2 (50 mM Tris pH 7.8, 1 M NaCl, 2 mM β-mercaptoethanol), followed by a final wash step with buffer A. His-tagged protein was eluted with a buffer B (50 mM Tris pH 7.8, 500 mM NaCl, 1 M imidazole, and 1 mM β-mercaptoethanol). The fractions containing the eluted protein were incubated with TEV protease overnight at 4 °C. The sample was dialyzed against buffer A at 4 °C and purified by a second Ni$^{2+}$ affinity step. Cleaved AtPHR1$^{280-360}$ from the flow-through of the column was concentrated to 3 mL and purified further by SEC on a Superdex 75 16/600 (GE Healthcare) equilibrated in 25 mM HEPES pH 7.0, 200 mM NaCl and 0.5 mM TCEP at a flow rate of 1 mL/min. Fractions containing AtPHR1$^{280-360}$ were pooled, concentrated to 0.5 mL, and the protein concentration was determined via Bradford assay (AppliChem). Samples were flash-frozen in liquid nitrogen and subsequently used for NMR spectroscopy experiments.

**Isothermal titration calorimetry**. All ITC experiments were performed at 25 °C using a MicroCal PEAQ-ITC system (Malvern Panalytical) equipped with a 200 μl sample cell and a 40 μl injection syringe. InsP$_7$ and InsP$_8$ were produced as described[47]. All proteins were dialysed against ITC buffer (20 mM HEPES pH 7.0,

200 mM NaCl) and PP-InsP ligands were diluted in ITC buffer prior to all measurements. A typical titration consisted of 15 injections, the protein concentrations in the syringe and in the cell are provided in the respective figure legend. Data were analyzed using the MicroCal PEAQ-ITC analysis software (v1.21).

**Crystallization and crystallographic data collection**. Two hexagonal crystal forms containing AtPHR1$^{280-360}$ only developed in sitting drops consisting of 0.2 μL protein at a concentration of 12 mg/mL and 0.2 μL reservoir solution (0.1 M phosphate citrate pH 4.2, 0.2 M NaCl, 20% (w/v) PEG 8000). Crystals were cryo-protected by adding reservoir solution containing 10% (v/v) ethylene glycol directly to the drop and subsequently snap-frozen in liquid nitrogen. A third, tetragonal crystal form developed in 0.1 M Bis-Tris pH 6.5, 0.1 M NaCl, 1.5 M (NH$_4$)$_2$SO$_4$. Crystals were cryo-protected by serial transfer into reservoir solution supplemented with 15% (v/v) glycerol and snap-frozen in liquid nitrogen. Crystal forms 1, 2, and 3 diffracted to ~2.4, ~2.5, and ~1.9 Å resolution, respectively. Data were collected at beam-line PXIII of the Swiss Light Source, Villigen, Switzerland. Data processing and scaling were done in XDS[48].

**Crystallographic structure solution and refinement**. The AtPHR1$^{280-360}$ structure was solved by molecular replacement using the previously described SPX$^{CtGde1}$ (PDB-ID:5IJJ) core helices as search model in calculations with the program PHASER[49]. The structure was completed in iterative cycles of manual model building in COOT[50] and restrained refinement in phenix.refine[51] or Refmac5[52]. Residues 280–294 and 278–280 appear disordered in the final model. Quality of the structural model was assessed by using MolProbity[53], refinement statistics are shown in Supplementary Table 5. Structure visualization was done with PyMOL (Molecular Graphics System, Version 1.8, Schrödinger, LLC) and ChimeraX[54]. The structure of AtSPX1 was modeled using the program SWISS-MODEL[55] and the SPX$^{HsXPR1}$ domain structure of the human phosphate exporter as template (PDB-ID:5IJH, GMQE score ~ 0.49, QMEAN4 score ~ −2.27, 29.5% sequence identity)[30]. Conserved PP-InsP-binding residues in AtSPX1 were determined by aligning sequences with previously described SPX domains[30] using the program T-coffee[56].

**Right-angle light scattering**. The oligomeric state of AtPHR1 variants was analyzed by SEC paired with a refractive index detector using an OMNISEC RESOLVE/REVEAL combined system (Malvern). Instrument calibration was performed with a bovine serum albumin (BSA) standard (Thermo Scientific Albumin Standard). Samples of 50 μL containing 2–10 mg/mL (wild-type AtPHR1$^{280-360}$, AtPHR1$^{280-360\ Olig1}$, AtPHR1$^{280-360\ Olig2}$, AtPHR1$^{280-360\ KHR/A}$, wild-type AtPHR1$^{222-358}$, AtPHR1$^{222-358\ Olig1}$, and AtPHR1$^{222-358\ Olig2}$) in OMNISEC buffer (20 mM HEPES pH 7.5, 150 mM NaCl) were separated on a Superdex 200 increase 10/300 GL column (GE Healthcare) at a column temperature of 25 °C and a flow rate of 0.7 ml min$^{-1}$. Data were analyzed using the OMNISEC software (v10.41).

**DNA oligonucleotide annealing**. DNA oligonucleotides were dissolved in annealing buffer (10 mM HEPES-NaOH pH 8.0, 50 mM NaCl, 0.1 mM EDTA). Equal volumes of the equimolar DNA oligonucleotides were mixed and incubated in a heat block for 5 min at 95 °C. Subsequently, DNA oligonucleotides were cooled down to room temperature for 90 min. Double-stranded DNA oligonucleotides were aliquoted and stored at −20 °C.

**Electrophoretic mobility shift assay**. Mini-PROTEAN TBE precast gels (5%; Bio-Rad) have been pre-electrophoresed in 0.5× TBE buffer for 60 min at 70 V. Reactions mixes have been prepared following the Odyssey® Infrared EMSA kit manual (LI-COR) without the use of optional components, including 50 nM of IRDye800 end-labeled oligos (refer to Supplementary Table 4a; Metabion), and a 1 : 5 dilution series of wild-type AtPHR1$^{222-358}$ or AtPHR1$^{222-358\ Olig1}$ (1.2 μg to 76.8 pg). Reaction mixes have been incubated for 30 min at room temperature in the dark and 2 μL of 10× Orange Loading Dye (LI-COR) have been added to each sample prior to loading on a 5% TBE gel. Gels have been electrophoresed until orange dye migrated to the bottom of the gel (~1 h) at 70 V in the dark. Gels have been scanned with the 800 nm channel of an Odyssey imaging system (LI-COR).

**Grating-coupled interferometry**. All GCI experiments were performed at 4 °C using a Creoptix WAVE system (Creoptix sensors) with 4PCP WAVE chips (Creoptix sensors). Chips were conditioned with borate buffer (100 mM sodium borate pH 9.0, 1 M NaCl) and subsequently neutravidin was immobilized on the chip surface via standard amine-coupling: activation (1 : 1 mix of 400 mM N-(3-dimethylaminopropyl)-N′-ethylcarbodiimide hydrochloride and 100 mM N-hydroxysuccinimide), immobilization (30 μg ml$^{-1}$ of neutravidin in 10 mM sodium acetate, pH 5.0), passivation (5% BSA in 10 mM sodium acetate pH 5.0), and quenching (1 M ethanolamine, pH 8.0). Biotinylated oligos (Supplementary Table 4b; Metabion) were captured on the chip. Analytes were injected in a 1 : 2 dilution series starting from 4 μM (AtPHR1$^{222-358}$), 20 μM (AtPHR1$^{223-358\ Olig1}$, AtPHR1$^{223-358\ Olig2}$), or 10 μM (OsPHR2, OsPHR2$^{KHR/A}$) in GCI buffer (for OsPHR2: 20 mM HEPES pH 7.9, 200 mM NaCl; for AtPHR1: 20 mM HEPES pH

7.5, 300 mM NaCl). Blank injections every fourth cycle were used for double referencing and a dimethylsulfoxide (DMSO) calibration curve (0%, 0.5%, 1%, 1.5%, 2%) for bulk correction. Data were corrected and analyzed using the Creoptix WAVE control software (corrections applied: X and Y offset, DMSO calibration, double referencing, refractive index correction), and a one-to-one binding model was used to fit all experiments.

**CD spectroscopy**. Far-UV CD spectra and melting curves were acquired with a Chirascan V100 CD spectrometer holding the protein solutions in quartz cuvettes of 1 mm optical path, scanning from 260 to 190 nm at 1 nm/s with a slit width of 1 nm against an air background obtained with the same settings without any cuvette. Protein concentration was 0.08 mg/mL (8.5 µM) in all experiments, the buffer was 10 mM KPi pH 7.5. A reference spectrum of the buffer showed only noise within ±0.5 mdeg. All the final spectra reported were obtained with a single scan, as we observed very good signal-to-noise and no saturation of the detector voltage even at low wavelengths. Melting curves were acquired with 2 °C stepwise increments and 30 s intervals from 24 to 98 °C. Data were fitted in the range from 195 nm to 250 nm using the three-component (α-helix, β-strand, and random coil) fitting (http://lucianoabriata.altervista.org/jsinscience/cd/cd3.html) as previously described[57].

**NMR spectroscopy**. TROSY spectra of [U-$^{15}$N,$^2$H]-AtPHR1$^{280-360}$ and [U-$^{15}$N,$^2$H]-AtPHR1$^{280-360}$ $^{KHR/A}$ were acquired on a 600 MHz spectrometer with a cryo-probe at 30 °C. The NMR buffer contained 25 mM HEPES pH 7.0, 200 mM NaCl and 0.5 mM TCEP. NMR data were analyzed by CCPNMR 2.4.2[58]. The dissociation constant $K_D$ was obtained by calculating chemical shift perturbations of each point by and plotted against InsP$_8$ concentration. The data points were fitted in MATLAB.

**Plant material, seed sterilization, and plant growth conditions**. All *A. thaliana* plants used in this study were of the Columbia (Col-0) ecotype. Seeds of the T-DNA insertion lines *phr1-3* (SALK_067629) and *phl1* (SAIL_731_B09) were obtained from the European Arabidopsis Stock Center. Homozygous *phr1-3* and *phl1* lines were identified by PCR using T-DNA left and right border primers paired with gene-specific sense and antisense primers (Supplementary Table 3d). The *phr1 phl1* double mutant was kindly provided by Yves Poirier (University of Lausanne, Switzerland); *vih1-2 vih2-4* double and *phr1 phl1 vih1-2 vih2-4* quadruple mutants have been reported previously[35]. Seeds were surface sterilized by incubation in 70% (v/v) CH$_3$-CH$_2$-OH for 10 min, followed by incubation in 0.5% (v/v) sodium hypochlorite for 10 min, and subsequently washed four times in sterile H$_2$O. Seeds were placed on full half-strength Murashige–Skoog plates[59] containing 1 (w/v) % sucrose and 0.8 (w/v) % agar ($^{1/2}$MS plates), and stratified for 2–3 days at 4 °C in the dark prior to transfer into a growth cabinet. Plants were grown on vertical $^{1/2}$MS plates at 22 °C under long day conditions (16 h light–8 h dark) for 8 to 11 days.

**Western blotting**. Proteins were transferred to nitrocellulose membrane (GE Healthcare, Amersham$^{TM}$ Highbond$^{TM}$-ECL) via wet western blotting at 4 °C and 30 V overnight. Membranes were blocked in TBS-Tween (0.1%)–Milk (5%) for 1 h at room temperature. For mCherry detection, membranes were incubated overnight with an anti-mCherry antibody (ab167453, dilution 1 : 2000; Abcam) followed by 1 h incubation with an anti-rabbit-horseradish peroxidase (HRP) antibody (dilution 1 : 10,000). For GFP detection, membranes were incubated overnight with an anti-GFP-HRP antibody (130-091-833, dilution 1 : 1000, Miltenyi Biotec). For FLAG detection, membranes were incubated overnight with an anti-FLAG-HRP antibody (A8692, dilution 1 : 1000, Sigma). For SPX1 detection, membranes were incubated overnight with an anti-SPX1 antibody (dilution 1 : 1000, kind gift of Professor Mingguang Lei) followed by one hour incubation with an anti-rabbit-HRP antibody (dilution 1 : 10,000, Calbiochem). Antibodies were diluted in TBS-Tween (0.05%)–Milk (2.5%). Membranes were detected with SuperSignal™ West Femto Maximum Sensitivity Substrate (34095, Thermo Scientific™).

**Determination of cellular Pi concentrations**. To determine cellular Pi concentration at seedling stage, plants were transferred from $^{1/2}$MS plates to -Pi $^{1/2}$MS plates containing 1 (w/v)% sucrose and 0.8 (w/v)% agarose supplemented with either 0, 1, or 10 mM Pi (KH$_2$PO$_4$/K$_2$HPO$_4$ pH 5.7) at 7 DAG, and grown at 22 °C under long day conditions. At 14 DAG, seedlings were weighted and harvested into 1.5 mL tubes containing 500 µL nanopure H$_2$O. Samples were frozen at −80 °C overnight, thawed at 80 °C for 10 min, refrozen at −80 °C, incubated at 80 °C in a thermomixer shaking at 1400 r.p.m. for 1 h, and briefly centrifuged to sediment plant tissue. Pi content was measured by the colorimetric molybdate assay[60]. In brief, 600 µL ammonium molybdate solution (0.44 g of ammonium molybdate tetra hydrate in 97.3 mL nanopure water; add 2.66 mL concentrated (18 M) H$_2$SO$_4$ to a final volume of 100 mL), 100 µL 10% ascorbic acid, and 300 µL sample (250 µL nanopure water + 50 µL extracts) or 300 µL NaPi standard solution were mixed. Samples were incubated at 37 °C for 1 h and absorbance at 820 nm was measured.

**RNA analyses**. At 14 DAG, 50–150 mg seedlings were harvested in 2 ml Eppendorf tubes containing two metal beads each, shock-frozen in liquid nitrogen and ground in a tissue lyzer (MM400, Retsch). RNA extraction was performed using the ReliaPrep RNA Tissue Miniprep System (Promega) including in column DNaseI treatment to remove genomic DNA. First strand cDNA synthesis was performed from 1 to 2.5 µg of total RNA using Superscript II RT (Invitrogen) with oligo(dT) primers. Quantitative reverse transcriptase PCR (qRT-PCR) was performed in 10 µL reactions containing 1× SYBR-Green fluorescent stain (Applied Biosystems) and measured using a 7900HT Fast Real Time PCR-System (Applied Biosystems). qRT-PCR program: 2'–95 °C; 40× (30"–95 °C; 30"–60 °C; 20"–72 °C); melting curve 95 °C–60 °C–95 °C. A primer list can be found in Supplementary Table 3e. Expression levels of target genes were normalized against the housekeeping gene *Actin2*. For every genotype, three biological replicates were analyzed in technical triplicates.

**Transient transformation of *N. benthamiana***. For each construct, 4 ml of *A. tumefaciens* strain pGV2260 suspension culture were grown overnight at 28 °C. Cells were collected by centrifugation at 700 × *g* for 15 min and resuspended in transformation buffer (10 mM MgCl$_2$, 10 mM MES pH 5.6, 150 µM acetosyringone). Cell density was measured and set to a final OD$_{600nm}$ of 0.5 for SPX1 and PHR1, and to 0.1 for the silencing suppressor P19. Suspension cultures were incubated for 2 h in the dark at room temperature and subsequently mixed at a volume ratio of 1 : 1 : 1 (SPX1 : PHR1 : P19). *N. benthamiana* leaves were infiltrated using a 0.5 ml syringe and 3 leaf disks (*d* = 1 cm) per sample were harvested after 3 days, snap-frozen in liquid nitrogen, and stored at −80 °C.

**Co-immunoprecipitation**. For co-immunoprecipitation experiments with proteins transiently expressed in *N. benthamiana*, samples were ground in liquid nitrogen with plastic mortars and proteins were extracted with 600 µL of homogenization buffer (50 mM Tris-HCl pH 7.5, 150 mM NaCl, 0.25% Triton X-100, 5% (v/v) glycerol, 1 mM phenylmethylsulfonyl fluoride, cOmplete$^{TM}$ EDTA-free protease inhibitor cocktail (Roche). Samples were incubated at 4 °C for 10 min with gently rotation and subsequently centrifuged at 16,000 × *g* and 4 °C for 15 min. Supernatants were transferred to fresh tubes and further centrifuged at 16,000 × *g* and 4 °C for 15 min. Supernatants were transferred to fresh tubes, while 50 µL of each supernatant were taken and mixed with 10 µL 6× SDS sample buffer (input), the remaining supernatants were mixed with 50 µL magnetic µMACS anti-GFP beads (Miltenyi Biotec) and incubated at 4 °C for 2 h with gently rotation. MACS columns (Miltenyi Biotec) were used with a µMACS Separator (Milteyi Biotec). MACS columns were washed with 200 µL of homogenization buffer and samples were loaded. Columns were washed either four times with 200 µL of homogenization buffer and once with wash buffer 2 (Miltenyi Biotec), or three times with 200 µL of homogenization buffer, three times with wash buffer 1 (Miltenyi Biotec) and once with wash buffer 2. Columns were incubated with 20 µL preheated elution buffer (Miltenyi Biotec) for 5 min at room temperature. Elution buffer (50 µL) was added and eluates were recovered. Inputs and eluates were boiled for 5 min at 95 °C prior and separated on 9% SDS-polyacrylamide electrophoresis gels. Co-immunoprecipitation experiments for proteins stably or natively expressed in *A. thaliana* were performed as previously described[29].

**Statistics**. All statistical analyses and data plots have been performed and generated with R version 3.5.2[61] and the packages "Hmisc[62]," "agricolae[63]," "car[64]," "multcompView[65]," and "multcomp[66]." qRT-PCR data were power transformed with the Box–Cox transformation and a significant one-way analysis of variance (ANOVA) followed by a Dunnett's post hoc test was performed for multiple comparisons of several genotypes vs. wild type (Col-0) shown in Supplementary Fig. 8a. Pi content was analyzed with a one-way ANOVA followed by a Dunnett's post hoc test for multiple comparisons of several genotypes vs. wild type (Col-0) shown in Fig. 4b and Supplementary Fig. 8b–d. (*$p < 0.5$; **$p < 0.01$; ***$p < 0.001$)

**Reporting summary**. Further information on research design is available in the Nature Research Reporting Summary linked to this article.

## Data availability

Data supporting the findings of this manuscript are available from the corresponding authors upon reasonable request. A reporting summary for this article is available as a Supplementary Information file. Coordinates and structure factors have been deposited in the Protein Data Bank (PDB) with accession codes 6TO5 (form1), 6TO9 (form2), and 6TOC (form 3). The associated X-ray diffraction images and data processing files have been deposited at http://zenodo.org with DOIs https://doi.org/10.5281/zenodo.3570698 (form1), https://doi.org/10.5281/zenodo.3570977 (form2), and https://doi.org/10.5281/zenodo.3571040 (form 3). Source data are provided with this paper.

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

## Acknowledgements

This work was supported by European Research Council Consolidator Grant 818696/INSPIRE (to M.H.), by Swiss National Foundation Sinergia Grant CRSII5_170925 (to M.H., S.H., and D.F.), and by an HHMI International Research Scholar Award (to M.H.). M.K.R. was supported by an EMBO long-term fellowship (ALTF-129-2017). We thank Irene Sabater for providing LI SPX1 and members of the Hothorn lab for critically reading the manuscript.

## Author contributions

M.K.R.: Conceptualization, data curation, formal analysis, validation, investigation, visualization, methodology, and writing (original draft, review, and editing). R.W.: Conceptualization, data curation, formal analysis, validation, investigation, methodology, and writing (review and editing). J.Z.: Investigation, methodology, and writing (review and editing). J.P.: Conceptualization, data curation, formal analysis, validation, investigation, methodology, and writing (review and editing). K.S.: investigation, formal analysis, and methodology. L.B.: Investigation and methodology. R.K.H.: Methodology. L.A.A.: formal analysis and methodology. L.A.H.: Software, formal analysis, and methodology. D.F.: Resources, methodology, and writing (review and editing). S.H.: Resources, methodology, supervision, and writing (review and editing). M.H.: Conceptualization, resources, data curation, formal analysis, supervision, funding acquisition, validation, investigation, visualization, methodology, project administration, and writing (original draft, review, and editing).

## Competing interests

The authors declare no competing interests.
