## [Peer Review File · Nature Communications]

Reviewers' comments:

Reviewer #1 (Remarks to the Author):

Phosphorus (Pi) is an essential building block for various functionally important molecules, such as phospholipids, nucleic acids, and nucleotides. Phosphate starvation response 1 protein (PHR1) and SPX domain protein play critical roles in Pi starvation signaling in Arabidopsis. PHR1 is a founding member of transcription factors containing both MYB and coiled-coil (CC) domains. In consistent with previous studies, Ried and the coworkers demonstrated that AtPHR1 interacts with AtSPX protein in Pi-dependent manner. In addition, they also showed that mutation of surface exposed residues K325, H328, or R335 of AtPHR1 CC domain all disrupted the interaction between AtPHR1 and AtSPX1, confirming that AtPHR1 CC domain plays a critical role in the binding to AtSPX. Although most of the results in the manuscript are sound, major concerns are from the oligomeric state of AtPHR1.

1) The authors solved the structures of stand-alone AtPHR1 CC domain in three different forms. All structures were refined to high quality and unraveled one four-stranded anti-parallel arrangement of AtPHR1 CC domain. Although the structure is interesting and novel, it does not agree with their own and many previous studies, which suggested that CC domain may mediate the dimerization of AtPHR1. The obvious difference raises the question about the biological relevance of the structure. As stated by the authors, they failed to assess the oligomeric state of full-length AtPHR1, due to the rapid degradation of the recombinant protein. Maybe the authors can test the oligomeric state of homologous proteins of AtPHR1, such as OsPHR2 described in this study and their previous study (Control of eukaryotic phosphate homeostasis by inositol polyphosphate sensor domains, Science, 2006).

2) Mutation of residues located on the tetramerization interface of AtPHR1 CC domain converted both AtPHR1222 – 358 and AtPHR1280–360 into stable monomer. Via GCI assays, they showed that mutated AtPHR1222 – 358 and AtPHR1280–360 have weaker DNA binding affinity, compared to the WT protein. However, no obvious differences were observed when analyzed by EMSA assay. In fact, the DNA binding affinity of AtPHR1222 – 358 Oligo 1 mutant protein is stronger than that of WT protein at lower concentrations (Figure 2C). Therefore, it is necessary to re-evaluate the results and conclusions.

3) Mutation of L319, I333, L337, L317, L327, and I341 altered the oligomeric state of the AtPHR1222 – 358. However, it is not clear whether mutation of these residues affected the folding of AtPHR1 CC domain, which could be checked by CD spectrum.

4) Besides DNA binding, it is also worthy to investigate the interactions between SPX and AtPHR1222 – 358 Oligo 1 (or AtPHR1222 – 358 Oligo 2), which could provide some implications for the functional importance of the oligomeric state of AtPHR1 CC domain.

Reviewer #2 (Remarks to the Author):

Ried, et al. use an outstanding combination of structural biology, biochemistry and genetics to identify and validate PHR1 loss-of-function mutations that alter SPX interaction and phosphate responsiveness in plants. How plants respond to phosphate is very intriguing even to a general audience and should be considered a very strong candidate for publication.

The InsP binding site on SPX was known, the authors have done an excellent job beginning to show that InsP8 might sit between two basic patches on SPX and PHR acting as "molecular glue". Using 3 new crystal structures, they identified PHR "KHR" motif responsible for interacting with SPX, showing the residues are required for full phosphate responsiveness in A.t., functional interaction and direct interaction with SPX in vitro, in yeast and in whole plants. They further show direct and functional interactions between PHR and SPX require the enzymes needed to make InsP8, and multiple orthogonal aspects of plant physiology are effected by mutation of the new interface, suggesting InsP8 as a key InsP readout for phosphate responsiveness. Thus it appears the paper needs only to firm up that InsP8 sits in the identified interface between PHR and SPX to

be complete.

Major concerns:

1. Line 137 states, "the SPX-PHR interaction is mediated by PP-InsPs", but the authors made an SPX-PHR complex from bacterial expressed proteins in Fig 1B. If PP-InsPs are required for SPX-PHR interaction, how can the complex form between bacterially expressed proteins? If the proteins come off SEC as a complex, clearly PP-InsPs are not required for SPX-PHR interaction in vitro. The InsP8 may stabilize or change dynamics of the interaction, consistent with Fig 1B, but this discrepancy needs resolution.

2. Assays throughout the work suggest InsP8 mediates the interaction between PHR and SPX. Although the "KHR" motif in PHR is identified to mediate interaction with SPX, not enough evidence is provided the KHR mutation prevents interaction through some other mechanism. The mutation of a transcription factor (PHR), or genetic loss of InsP8 enzymes, can induce many changes in cellular proteome, RNAs, or other metabolites which could be responsible for altered SPX interactions in cells. Thus whether or not InsP8 mediates PHR-SPX interaction as molecular glue is at its core a structural question. The authors need to find a way to better show InsP binds PHR at the KHR site.

The small PHR constructs lend themselves to NMR without assigning, InsP8 should induce shifts or broadening in WT, less in KHR mutant (Fig 3 control R->A mutants of PHR1 could be controls). Side chains of KHR might change +/- InsP, should lose side chain peaks in mutant so no need to assign to unequivocally ID. Alternatively, a putative InsP derivative (spin or isotope) that could induce nmr changes would be expected to do so in WT, less so in KHR mutant of PHR. A solid InsP8-dependent structural change in PHR that influences WT, but not KHR mutant is needed.

Minor concerns:

1. Although *vih1/2* loss is phenocopied by KHR mutant complementation in the *phr1/phl1* null background, the role of *vih1/2* enzyme activity is not addressed (also for yeast *vip1/ kcs* genes). Since KHR mutant is putatively decoupled from InsP8 metabolism, any PHR-dependent phenotype in a background complemented with a KHR mutant allele should be somewhat independent of 1) *vih1/2* loss or 2) complementation with either wt or kinase-dead alleles of *vih1/2*. Some effort should be made to show that a KHR mutant phenotype is independent of the kinase activity of these genes.

2. Fig 2 convincingly demonstrates the oligomeric state of PHR1 is sensitive to mutations based on the novel structures presented in the paper. However, these studies also show the PHR1 tetrameric structure presented in Fig 1 is not present in constructs containing the MYB DNA binding domain, arguably the more relevant construct. The tetrameric structure is very valuable and is reasonably considered in discussion, but that the tetramer appears only in the most truncated construct needs to be better highlighted up front in the results.

Reviewer #3 (Remarks to the Author):

In this manuscript, Ried et al., report on a structure function analysis of the interaction between PHR1 and SPX1, two key players in phosphate (Pi) starvation signalling in plants. PHR1 is the master regulator of Pi starvation responses and SPX is the Pi sensor acting via a phosphate rich compound (Inositol pyrophosphate 8, InsP8). Using a Y2H approach, the authors mapped the SPX1 interaction domain of PHR1 to the presumed coiled coil (CC) domain. They determined the crystal structure of this interaction domain and found that in isolation it forms a tetramer

displaying an unusual four stranded antiparallel CC domain structure. Tetramerization of this domain is also observed in solution, but in the presence of the MYB DNA binding domain, they find the CC domain forms a dimer in solution. From the structure of the CC domain, they predict specific residues critical for dimerization and those critical for tetramerization. However, mutation of either type of residues renders the CC domain monomeric in solution (independent of whether the CC domain is examined in the presence or absence of the MYB domain). Moreover, mutation of residues critical for dimerization has a similar impact on DNA binding than mutation of residues critical for tetramerization. Altogether, this set of experiments reveal unexpected properties of the coiled coil domain of PHR1, although raises issues for clarification as mentioned below.

One important outcome of this study is that the structure of the CC domain allowed the authors to predict critical residues for the interaction with SPX1. Nicely, mutation of these residues impacted PHR1 binding to SPX but not to its DNA site and, in line with this, transgenic expression of mutants in these residues affects Pi homeostasis and the expression of Pi starvation induced genes.

Overall, this structure function study is technically sound and the conclusions on important residues for the PHR1-SPX interaction and the impact of their mutation are well established both in the biochemical and physiological context of PHR1 action. In addition, as suggested by the authors, the high conservation of the residues important for the interaction of PHR1 with SPX raises the possibility that SPX proteins could regulate a large proportion of members of this MYB-CC family, including APL that controls phloem development. This is a very interesting possibility, which is not experimentally explored in this study. In my opinion, a Y2H study of the interaction of selected subsets of SPX and PHR1-related proteins covering different phylogenetic distances could provide hints on the possibility that SPX action extends beyond the PHR1 subfamily. This would broaden the scope and increase the interest of this study

Additional comments:

1) I miss an explanation of why mutation of residues predicted to be involved in tetramerization of the CC domain affects dimerization of this domain; in the context of the previously described structure of PHR1 (Jiang et al 2019), what is the implication of oligo 2 residues in dimerization? In addition, for the sake of clarification, inclusion of a comment to explain the difference between the results shown in Figure 2 of the EMSA and the GCI experiment regarding DNA binding affinity of wild type PHR1 and mutants is also required. It is also unclear the behaviour of the PHR1oligo2 mutant in EMSA. Is it similar to that of the PHR1oligo 1 mutant?

2) The authors found that the binding affinity of PHR1-SPX to InsP8 is only around two-fold higher than that to InsP7. I agree with the authors this is quite a small difference to explain the preference for InsP8 in vivo. I notice however that the authors have examined the relative binding affinity of these inositol pyrophosphates using SPX4 and PHR2. The SPX4-PHR2 couple might display non-canonical behaviour because Pi also regulates SPX4 stability. I would like to see examination of the relative binding affinities of InsP8 and InsP7 to the PHR1-SPX1 complex

Ried & Wild et al., RESPONSE TO REVIEWERS' COMMENTS

Reviewer #1 (Remarks to the Author):

The revised manuscript is much improved and the authors have addressed my comments.

Reviewer #2 (Remarks to the Author):

The authors have addressed the stated concerns for the most part, if they qualify conclusions from Supplemental 6 the data will match the conclusions and are suitable for publication.

Rev 2 Major Concerns:

1. This concern was completely addressed by clarifying previously published data to this reviewer, and by changes the authors made to methods.

2A. This concern was only partially addressed but could be fixed by adding a sentence. The authors state in rebuttal they do not claim InsPs bind directly to the KHR motif in PHR1. The problem is that the language they use can easily be misinterpreted:

New Line 216: “A similar set of surface exposed basic residues has been previously found to form the binding site for PP-InsPs in various SPX receptors”... .. they go on to conclude... New Line 237: “Taken together, three highly conserved basic residues located at the surface of the PHR CC domain are critical for the interaction with the PP-InsP bound SPX receptor (Supplementary Fig. 4C) but seem not to be directly involved in low affinity InsP8 binding by the isolated CC domain (Supplementary Fig 6).” This language does not clearly state what the new NMR data begin to suggest: “If PHR1 binds specifically to InsP8 at all, it does so with very poor affinity, as measured by NMR using the isolated Arabidopsis CC domain”, which should be added to the text. Further critique of the NMR experimental design are below.

OUR RESPONSE: We thank the reviewer for pointing this out. As suggested, we have revised our conclusion from these experiments. The revised statement reads (lines 237-239): “The NMR titration experiments suggest that AtPHR1 does not contribute to the specific binding of InsP₈ and that the low affinity interaction between the CC domain and InsP₈ does not involve the KHR motif (Supplementary Fig. 6).”

2B. The new NMR data were poorly controlled requiring a qualifying sentence. IP6 (not potassium phosphate) should have been used as negative control since IP6 has no pyro group which was the subject of the inquiry. If the authors understandably don't want to repeat the NMR the following sentence should be added: “It remains possible the weak InsP8-Phr1 binding events we observed may occur independent of inositol pyrophosphorylation, as it is unknown if PHR1 binds inositol phosphates such as IP6 at similarly high concentrations.”

OUR RESPONSE: We fully agree with the reviewer that the isolated AtPHR1 CC domain does not contribute to the specific recognition of InsP₈ and in fact may not at all provide a binding site for this molecule. We have accordingly revised the discussion section of our manuscript which now reads (lines 327-329): “The newly identified basic surface area in PHR CC, harboring the conserved KHR motif, likely forms part of the SPX – PHR complex interface (Fig. 3a).” We would like to note, that we chose Pi as a control ligand in our NMR titrations as previous studies had claimed that Pi and not InsP₈ is the nutrient messenger promoting the association of SPX and PHR1 (Puga et al., PNAS, 2014; Wang et al., PNAS, 2014). This view has later been corrected by showing that SPX domains are specific receptors for PP-InsPs and not Pi (Wild et al., Science, 2016) and that the InsP8 generating PPIP5Ks VIH1 and VIH2 act upstream of SPX-PHR1 (Zhu et al., eLife, 2019; Dong et al., Mol Plant, 2019).

Rev 2 Minor concerns:

1. The new NMR data suggesting the KHR site does NOT directly interact with IP8 (supplemental 6) makes this experiment moot regardless of COVID, so this concern was addressed indirectly.
2. This concern was addressed by showing the homolog PHR2 can be a tetramer by light scattering, suggesting PHR1 could also exist as a tetramer at least in this orthogonal experiment.

Reviewer #3 (Remarks to the Author):

Ried et al., have made an important effort to deal with the reviewer's comments and criticisms, and in my opinion the manuscript is now suitable for publication. I think the Y2H data showing that, in addition to PHR1, SPX proteins interact with other members of the MYC-CC family is pertinent in the context of this study. Clearly further work will be necessary to understand why there are several MYB-CC members that do not interact with SPX proteins in the Y2H assay, despite having the conserved KRH motif, but this should be the subject of a separate study.

REVIEWERS' COMMENTS

Reviewer #1 (Remarks to the Author):

The revised manuscript is much improved and the authors have addressed my comments.

Reviewer #2 (Remarks to the Author):

The authors have addressed the stated concerns for the most part, if they qualify conclusions from Supplemental 6 the data will match the conclusions and are suitable for publication.

Rev 2 Major Concerns:

1. This concern was completely addressed by clarifying previously published data to this reviewer, and by changes the authors made to methods.

2A. This concern was only partially addressed but could be fixed by adding a sentence. The authors state in rebuttal they do not claim InsPs bind directly to the KHR motif in PHR1. The problem is that the language they use can easily be misinterpreted:

New Line 216: "A similar set of surface exposed basic residues has been previously found to form the binding site for PP-InsPs in various SPX receptors"...

... they go on to conclude...

New Line 237: "Taken together, three highly conserved basic residues located at the surface of the PHR CC domain are critical for the interaction with the PP-InsP bound SPX receptor (Supplementary Fig. 4C) but seem not to be directly involved in low affinity InsP8 binding by the isolated CC domain (Supplementary Fig 6)."

This language does not clearly state what the new NMR data begin to suggest: "If PHR1 binds specifically to InsP8 at all, it does so with very poor affinity, as measured by NMR using the isolated Arabidopsis CC domain", which should be added to the text. Further critique of the NMR experimental design are below.

2B. The new NMR data were poorly controlled requiring a qualifying sentence. IP6 (not potassium phosphate) should have been used as negative control since IP6 has no pyro group which was the subject of the inquiry. If the authors understandably don't want to repeat the NMR the following sentence should be added: "It remains possible the weak InsP8-Phr1 binding events we observed may occur independent of inositol pyrophosphorylation, as it is unknown if PHR1 binds inositol phosphates such as IP6 at similarly high concentrations."

Rev 2 Minor concerns:

1. The new NMR data suggesting the KHR site does NOT directly interact with IP8 (supplemental 6) makes this experiment moot regardless of COVID, so this concern was addressed indirectly.

2. This concern was addressed by showing the homolog PHR2 can be a tetramer by light scattering, suggesting PHR1 could also exist as a tetramer at least in this orthogonal experiment.

Reviewer #3 (Remarks to the Author):

Ried et al., have made an important effort to deal with the reviewer's comments and criticisms, and in my opinion the manuscript is now suitable for publication. I think the Y2H data showing that, in addition to PHR1, SPX proteins interact with other members of the MYC-CC family is pertinent in the context of this study. Clearly further work will be necessary to understand why there are several MYB-CC members that do not interact with SPX proteins in the Y2H assay, despite having the conserved KRH motif, but this should be the subject of a separate study.

Reviewer #1 (Remarks to the Author):

Phosphorus (Pi) is an essential building block for various functionally important molecules, such as phospholipids, nucleic acids, and nucleotides. Phosphate starvation response 1 protein (PHR1) and SPX domain protein play critical roles in Pi starvation signaling in Arabidopsis. PHR1 is a founding member of transcription factors containing both MYB and coiled-coil (CC) domains. In consistent with previous studies, Ried and the coworkers demonstrated that AtPHR1 interacts with AtSPX protein in Pi-dependent manner. In addition, they also showed that mutation of surface exposed residues K325, H328, or R335 of AtPHR1 CC domain all disrupted the interaction between AtPHR1 and AtSPX1, confirming that AtPHR1 CC domain plays a critical role in the binding to AtSPX. Although most of the results in the manuscript are sound, major concerns are from the oligomeric state of AtPHR1.

1) The authors solved the structures of stand-alone AtPHR1 CC domain in three different forms. All structures were refined to high quality and unraveled one four-stranded anti-parallel arrangement of AtPHR1 CC domain. Although the structure is interesting and novel, it does not agree with their own and many previous studies, which suggested that CC domain may mediate the dimerization of AtPHR1. The obvious difference raises the question about the biological relevance of the structure. As stated by the authors, they failed to assess the oligomeric state of full-length AtPHR1, due to the rapid degradation of the recombinant protein. Maybe the authors can test the oligomeric state of homologous proteins of AtPHR1, such as OsPHR2 described in this study and their previous study (Control of eukaryotic phosphate homeostasis by inositol polyphosphate sensor domains, Science, 2006).

OUR RESPONSE: We thank the reviewer for suggesting this experiment. We indeed find that full-length OsPHR2 expressed and purified from *E. coli* migrates as a tetramer in SEC-RALS experiments (newly added Supplementary Fig. 3). We have revised our statement in the results section accordingly (lines 168-172): “While we found purified full-length AtPHR1 to be too unstable for SEC-RALS analysis, full-length OsPHR2 behaved as a tetramer in solution (Supplementary Fig. 3). In contrast, untagged AtPHR1^{222 – 358}, which comprises the CC and the MYB DNA-binding domains only, runs as a dimer (Fig. 2a,b; black traces), in agreement with earlier reports².” We have updated our discussion section accordingly (lines 287-291): “Our AtPHR1 MYB-CC construct behaves as a dimer in solution, consistent with the recently reported crystal structure of the AtPHR1 MYB – DNA complex, and with earlier reports^{39,2}. Purified full-length OsPHR2 however appears to be a homotetramer in solution (Supplementary Fig. 3).”

2) Mutation of residues located on the tetramerization interface of AtPHR1 CC domain converted both AtPHR1^{222 – 358} and AtPHR1^{1280–360} into stable monomer. Via GCI assays, they showed that mutated AtPHR1^{222 – 358} and AtPHR1^{1280–360} have weaker DNA binding affinity, compared to the WT protein. However, no obvious differences were observed when analyzed by EMSA assay. In fact, the DNA binding affinity of AtPHR1^{222 – 358} Oligo 1 mutant protein is stronger than that of WT protein at lower concentrations (Figure 2C). Therefore, it is necessary to re-evaluate the results and conclusions.

OUR RESPONSE: We agree with the reviewer that in the EMSA assay the bands corresponding to AtPHR1^{222 – 358} Olig1 appear to be stronger than compared to wild-type AtPHR1^{222 – 358}. However, it is very difficult to conclude binding affinities/kinetics from the qualitative EMSA assay. This is why we established the quantitative GCI-based protein-DNA interaction assay in our lab and optimized it for AtPHR1-DNA binding. In our experience, AtPHR1^{222 – 358} Olig1 and AtPHR1^{222 – 358} Olig2 are more stable than wild-type AtPHR1^{222 – 358} in the buffer used for the EMSA experiments (Odyssey EMSA standard buffer for 30 minutes at RT). The GCI experiments were performed with different buffers at 4 °C, and we consistently found that wild-type AtPHR1^{222 – 358} had 10 – 20 fold higher affinity to its DNA binding motif when compared to AtPHR1^{222 – 358} Olig1/Olig 2. Moreover, it is

apparent from our GCI experiments that the binding kinetics of AtPHR1^{222 – 358} Olig1/Olig2 (fast association, fast dissociation) are very different from what we observe for the wild-type MYB-CC fragment (fast association, slow dissociation). We would like to include both experiments in our revised manuscript to highlight that the monomeric mutants of AtPHR1 can still bind to the promoter fragment, but with reduced binding affinity and with different binding kinetics.

3) Mutation of L319, I333, L337, L317, L327, and I341 altered the oligomeric state of the AtPHR222 – 358. However, it is not clear whether mutation of these residues affected the folding of AtPHR CC domain, which could be checked by CD spectrum.

OUR RESPONSE: We have performed the suggested circular dichroism experiments for wild-type AtPHR1280-360 and for the corresponding Olig1 and Olig2 mutant proteins. The data are shown in newly added Supplementary Figure 5. A new statement describing these experiments has been added to the results section (lines 191-200): “Analysis of the secondary structure content of wild-type AtPHR²⁸⁰⁻³⁶⁰ using circular dichroism (CD) spectroscopy revealed a 100% α -helical protein (Supplementary Fig. 5a), in agreement with our structural model (Fig. 1c). In contrast, we estimated the secondary structure content of the Olig1 and Olig2 mutant proteins to be ~50% α -helical and ~50 random coil (Supplementary Fig. 5a). The CD melting spectrum for wild-type AtPHR²⁸⁰⁻³⁶⁰ indicated the presence of a well-folded protein with a melting temperature (T_m) of ~ 50 °C, while could not reliably determine T_m 's for the Olig1 and Olig2 mutant proteins (Supplementary Fig. 5b). We conclude that mutation of either AtPHR1^{L319}, AtPHR1^{I333}, AtPHR1^{L337}, or AtPHR1^{L317}, AtPHR1^{L327}, AtPHR1^{I341} to asparagine disrupts the tetrameric coiled-coil domain of AtPHR1 and affects the structural integrity of the contributing α -helix.” Kristina Sturm and Luciano A. Abriata performed and analyzed the CD experiments shown in Supplementary Fig. 5 and have been included as authors in our revised manuscript. The methods section has been updated accordingly.

4) Besides DNA binding, it is also worthy to investigate the interactions between SPX and AtPHR1222 – 358 Oligo 1 (or AtPHR1222 – 358 Oligo 2), which could provide some implications for the functional importance of the oligomeric state of AtPHR1 CC domain.

OUR RESPONSE: From the CD spectroscopy analysis described in response to reviewer #1 point #3 (see above), we conclude that the α -helix contributing to the formation of the AtPHR1 coiled-coil domain and harboring the residues required for the interaction with SPX domains is at least partially unfolded at 25 deg C. Since we perform our ITC and GCI PHR – SPX binding assays at 25 deg C or above, we feel that the experiment suggested by the reviewer would be very difficult to interpret.

Reviewer #2 (Remarks to the Author):

Ried, et al. use an outstanding combination of structural biology, biochemistry and genetics to identify and validate PHR1 loss-of-function mutations that alter SPX interaction and phosphate responsiveness in plants. How plants respond to phosphate is very intriguing even to a general audience and should be considered a very strong candidate for publication.

The InsP binding site on SPX was known, the authors have done an excellent job beginning to show that InsP8 might sit between two basic patches on SPX and PHR acting as "molecular glue". Using 3 new crystal structures, they identified PHR “KHR” motif responsible for interacting with SPX, showing the residues are required for full phosphate responsiveness in A.t., functional interaction and direct interaction with SPX in vitro, in yeast and in whole plants. They further show direct and functional interactions between PHR and SPX require the enzymes needed to make InsP8, and multiple orthogonal aspects of plant physiology are effected by mutation of the new interface, suggesting InsP8 as a key InsP readout for phosphate responsiveness. Thus it appears the paper

needs only to firm up that InsP8 sits in the identified interface between PHR and SPX to be complete.

Major concerns:

1. Line 137 states, “the SPX-PHR interaction is mediated by PP-InsPs”, but the authors made an SPX-PHR complex from bacterial expressed proteins in Fig 1B. If PP-InsPs are required for SPX-PHR interaction, how can the complex form between bacterially expressed proteins? If the proteins come off SEC as a complex, clearly PP-InsPs are not required for SPX-PHR interaction *in vitro*. The InsP8 may stabilize or change dynamics of the interaction, consistent with Fig 1B, but this discrepancy needs resolution.

OUR RESPONSE: We are sorry for having caused this misunderstanding, which might have originated from a poorly phrased methods section. The statement in line 137 is correct, we only observe interaction between plant stand-alone SPX domains and PHR transcription factors in the presence of PP-InsPs (Wild et al., *Science*, 2016; see Figure 4b from this work below for your reference). In Fig. 1b of the current manuscript, we assess the binding of InsP₇ and InsP₈ to a 1:1 mixture of OsSPX4 and OsPHR2, which in the presence of the ligand will form a complex with both proteins likely contributing to the formation of the PP-InsP binding site.

In the methods subsections “Protein expression and purification” and “Crystallization and crystallographic data collection” we described our

attempt to purify and crystallize a AtSPX1 – PP-InsP – AtPHR1 complex. We later found that bacterial expressed AtSPX1 and the somewhat better behaving AtSPX2 are largely unfolded (see our response to point #2 of reviewer #3 below) and thus unable to bind PP-InsPs and to form a complex with AtPHR1 *in vitro*. This is why crystallization of this putative complex yielded a structure of the isolated AtPHR1 CC fragment. We thank the reviewer for bringing this issue to our attention. The revised method statement reads (lines 443-444): “For the purification of a putative AtSPX1 – InsP₈ - AtPHR1²⁸⁰⁻³⁶⁰ complex, the AtPHR1 cell pellet was thawed...” (lines 460-461): “Fractions containing the co-eluting AtSPX1¹⁻²⁵¹ and AtPHR1²⁸⁰⁻³⁶⁰ proteins...” (lines 463 – 465): “It was later found that AtSPX1¹⁻²⁵¹ expressed in *E. coli* is largely unfolded and unable to bind InsP₈ and hence no complex with AtPHR1²⁸⁰⁻³⁶⁰ was observed in our crystals (see below). “ (line 506): “Two hexagonal crystal forms containing AtPHR1²⁸⁰⁻³⁶⁰ only developed in sitting drops...”

2. Assays throughout the work suggest InsP8 mediates the interaction between PHR and SPX. Although the “KHR” motif in PHR is identified to mediate interaction with SPX, not enough evidence is provided the KHR mutation prevents interaction through some other mechanism. The mutation of a transcription factor (PHR), or genetic loss of InsP8 enzymes, can induce many changes in cellular proteome, RNAs, or other metabolites which could be responsible for altered SPX interactions in cells. Thus whether or not InsP8 mediates PHR-SPX interaction as molecular glue is at its core a structural question. The authors need to find a way to better show InsP binds PHR at the KHR site.

Response Fig. 1. OsSPX4 and OsPHR2 only for a complex in the presence of PP-InsPs. Reproduced from Wild et al., *Science*, 2016.

OUR RESPONSE: We do not, to the best of our knowledge, claim in our manuscript that PP-InsPs bind directly to the KHR motif in AtPHR1.

The small PHR constructs lend themselves to NMR without assigning, InsP₈ should induce shifts or broadening in WT, less in KHR mutant (Fig 3 control R->A mutants of PHR1 could be controls). Side chains of KHR might change +/- InsP, should lose side chain peaks in mutant so no need to assign to unequivocally ID. Alternatively, a putative InsP derivative (spin or isotope) that could induce nmr changes would be expected to do so in WT, less so in KHR mutant of PHR. A solid InsP₈-dependent structural change in PHR that influences WT, but not KHR mutant is needed.

OUR RESPONSE: Joka Pipercevic and Sebastian Hiller from the Biozentrum Basel, Switzerland have performed the NMR titration experiments suggested by reviewer #2. The experiments reveal a specific (potassium phosphate was used as control) but very weak interaction between wild-type ¹⁵N, ²H AtPHR1²⁸⁰⁻³⁶⁰ and InsP₈ with an estimated dissociation constant of ~ 2 mM (newly added Supplementary Fig. 6). The AtPHR1²⁸⁰⁻³⁶⁰ KHR/A mutant protein behaves very similar to wild type The new experiments are discussed in the text (lines 224-238): “Using nuclear magnetic resonance (NMR) spectroscopy, we next tested if the KHR motif in AtPHR1 is directly involved in PP-InsP ligand recognition. We titrated InsP₈ into ¹⁵N, ²H labeled AtPHR1²⁸⁰⁻³⁶⁰ and recorded TROSY spectra using potassium phosphate (KPi) as control. Five backbone amide moieties exhibited chemical shift perturbations in the presence of InsP₈ but not in the presence of the KPi control. We acquired titration spectra using increasing concentrations of InsP₈ and estimated dissociation constants for InsP₈ based on three representative peaks (Supplementary Fig. 6). The derived dissociation constant is in the millimolar range, and saturation could not be reached in the available concentration range (Supplementary Fig. 6). We next repeated the same set of experiments using the AtPHR1^{KHR/A} mutant protein, located the same peaks in the TROSY spectra and estimated similar dissociation constants (Supplementary Fig. 6).

Taken together, three highly conserved basic residues located at the surface of the PHR coiled-coil domain are critical for the interaction with the PP-InsP bound SPX receptor (Supplementary Fig. 4c) but seem not to be directly involved in low affinity InsP₈ binding by the isolated CC domain (Supplementary Fig. 6).”

Minor concerns:

1. Although *vih1/2* loss is phenocopied by KHR mutant complementation in the *phr1/phl1* null background, the role of *vih1/2* enzyme activity is not addressed (also for yeast *vip1/ kcs* genes). Since KHR mutant is putatively decoupled from InsP₈ metabolism, any PHR-dependent phenotype in a background complemented with a KHR mutant allele should be somewhat independent of 1) *vih1/2* loss or 2) complementation with either wt or kinase-dead alleles of *vih1/2*. Some effort should be made to show that a KHR mutant phenotype is independent of the kinase activity of these genes.

OUR RESPONSE: We have previously shown that the Arabidopsis diphosphoinositol pentakisphosphate kinases VIH1 and VIH2 redundantly control the biosynthesis of the signaling molecule InsP₈. *vih1/vih2* loss-of-function mutants lack InsP₈, display a seedling lethal phenotype and exhibit constitutive phosphate starvation responses. Additional deletion of the transcription factors PHR1 and PHL1 can partially suppress *vih1/vih2* mutant phenotypes, placing VIH1/2 in a common signaling pathway with PHR1 and suggesting that InsP₈ is the signaling molecule promoting the interaction between PHR1 and SPX domains *in planta* (Zhu *et al.*, *eLife*, 2019; Lorenzo-Orts *et al.*, *New Phytol.*, 2020). Indeed Dong *et al.*, *Mol Plant*, 2019 have demonstrated that the interaction between AtSPX1 and AtPHR1 is lost in the *vih1/vih2* mutant. We thus feel that the genetic interaction between VIHs and PHR1 is already well established.

We have nevertheless tried to perform the experiment suggested by the reviewer, by crossing a *vih1-2/vih2-4* +/- mutant with a *phr1/phl1* + PHR1^{KHR} mutant line (both homozygous *vih1-2/vih2-4* and *phr1* + PHR1^{KHR} mutants are seedling lethal, compare Fig. 4a). However, during in the course

of the COVID19 pandemic our growth facilities were shut down and we could not access our experiments. We thus apologize for not having been able to complete the suggested experiment.

2. Fig 2 convincingly demonstrates the oligomeric state of PHR1 is sensitive to mutations based on the novel structures presented in the paper. However, these studies also show the PHR1 tetrameric structure presented in Fig 1 is not present in constructs containing the MYB DNA binding domain, arguably the more relevant construct. The tetrameric structure is very valuable and is reasonably considered in discussion, but that the tetramer appears only in the most truncated construct needs to be better highlighted up front in the results.

OUR RESPONSE: This point has also been raised by reviewer #1 in point #1. We have experimentally addressed this issue, please see above.

Reviewer #3 (Remarks to the Author):

In this manuscript, Ried et al., report on a structure function analysis of the interaction between PHR1 and SPX1, two key players in phosphate (Pi) starvation signalling in plants. PHR1 is the master regulator of Pi starvation responses and SPX is the Pi sensor acting via a phosphate rich compound (Inositol pyrophosphate 8, InsP8). Using a Y2H approach, the authors mapped the SPX1 interaction domain of PHR1 to the presumed coiled coil (CC) domain. They determined the crystal structure of this interaction domain and found that in isolation it forms a tetramer displaying an unusual four stranded antiparallel CC domain structure. Tetramerization of this domain is also observed in solution, but in the presence of the MYB DNA binding domain, they find the CC domain forms a dimer in solution. From the structure of the CC domain, they predict specific residues critical for dimerization and those critical for tetramerization. However, mutation of either type of residues renders the CC domain monomeric in solution (independent of whether the CC domain is examined in the presence or absence of the MYB domain). Moreover, mutation of residues critical for dimerization has a similar impact on DNA binding than mutation of residues critical for tetramerization. Altogether, this set of experiments reveal unexpected properties of the coiled coil domain of PHR1, although raises issues for clarification as mentioned below.

One important outcome of this study is that the structure of the CC domain allowed the authors to predict critical residues for the interaction with SPX1. Nicely, mutation of these residues impacted PHR1 binding to SPX but not to its DNA site and, in line with this, transgenic expression of mutants in these residues affects Pi homeostasis and the expression of Pi starvation induced genes.

Overall, this structure function study is technically sound and the conclusions on important residues for the PHR1-SPX interaction and the impact of their mutation are well established both in the biochemical and physiological context of PHR1 action. In addition, as suggested by the authors, the high conservation of the residues important for the interaction of PHR1 with SPX raises the possibility that SPX proteins could regulate a large proportion of members of this MYB-CC family, including APL that controls phloem development. This is a very interesting possibility, which is not experimentally explored in this study. In my opinion, a Y2H study of the interaction of selected subsets of SPX and PHR1-related proteins covering different phylogenetic distances could provide hints on the possibility that SPX action extends beyond the PHR1 subfamily. This would broaden the scope and increase the interest of this study.

OUR RESPONSE: We have performed the suggested Y2H screen by testing binary interactions between the 15 MYB-CC family members and AtSPX1, 2, 3 and 4. We found robust interaction between SPX1/2/3/4 domain and AtPHR1, AtPHL1 and AtPHL5, some weaker interactions for other PHLs, but not detectable interaction with APL (At1g79430, WDY, PHL14). We currently don't understand why not all MYB-CCs interact with SPX domains, despite sharing a very conserved CC core and the residues required for interaction with SPX domains (compare

Supplementary Fig. 4c). A postdoctoral fellow is currently investigating the regulation of other MYB-CCs by SPX domain-containing proteins in Arabidopsis, but we find the present data to be too preliminary to be included in the manuscript. We would leave it to the editor to decide if the Y2H data shown in Response Fig. 2 should become part of the manuscript as a supplementary figure.

Response Fig. 2. Yeast-2-hybrid analysis of AtSPX1 vs. different MYB-CC family members from Arabidopsis.

Additional comments:

1) I miss an explanation of why mutation of residues predicted to be involved in tetramerization of the CC domain affects dimerization of this domain; in the context of the previously described structure of PHR1 (Jiang et al 2019), what is the implication of oligo 2 residues in dimerization?

OUR RESPONSE: Please see points #1 raised by reviewer #1 above. Basically, we find that full-length OsPHR2 is a tetramer in solution (newly added Supplementary Fig. 3) as is the CC core domain. A fragment containing both the MYB and the CC domains behaves as a dimer in solution. Both the Olig1 and Olig2 mutation disrupt the CC domain, leading to a monomeric state that can still bind to DNA, but less tightly (compare Fig. 2). Based on our analyses AtPHR1 could be a homo- or hetero-tetramer in solution and we have revised our discussion section accordingly (lines 301-305): “An attractive hypothesis would thus be that AtPHR1 binds its target promoter as a dimer, but can potentially form homo-tetramers, or hetero-tetramers with other MYB-CC type transcription factors sharing the conserved, plant-unique CC structure and sequence (Supplementary Fig. 4). Notably, PHR1 PHL1 heteromers have been previously described⁷.”

In addition, for the sake of clarification, inclusion of a comment to explain the difference between the results shown in Figure 2 of the EMSA and the GCI experiment regarding DNA binding affinity of wild type PHR1 and mutants is also required. It is also unclear the behaviour of the PHR1oligo2 mutant in EMSA. Is it similar to that of the PHR1oligo 1 mutant?

OUR RESPONSE: Please refer to our response to point #2 raised by reviewer #1. The Olig1 and 2 mutants behave similar in EMSA assays.

2) The authors found that the binding affinity of PHR1-SPX to InsP8 is only around two-fold higher than that to InsP7. I agree with the authors this is quite a small difference to explain the preference for InsP8 in vivo. I notice however that the authors have examined the relative binding affinity of these inositol pyrophosphates using SPX4 and PHR2. The SPX4-PHR2 couple might display non-canonical behaviour because Pi also regulates SPX4 stability. I would like to see examination of the relative binding affinities of InsP8 and InsP7 to the PHR1-SPX1 complex.

OUR RESPONSE: We have tried to recombinantly express and purify various fragments of AtSPX1, AtSPX2, AtSPX3 and AtSPX4 for biochemical and structural studies in *E. coli* and in insect cells. The best behaving constructs were from AtSPX2 (amino-acid fragments 1-192, 1-234, 1-254 and 1-287) which we can nicely purify (Response Fig. 3). However, 2D NMR spectra recorded for all four constructs suggest that also AtSPX2 is only partially folded (Response Fig. 4). We nevertheless performed the experiment suggested by the reviewer using NMR spectroscopy. We found that neither addition of InsP₆ or of InsP₆/AtPHR1²⁸⁰⁻³⁶⁰ did induce chemical shift perturbations (Response Fig. 5). We thus cannot demonstrate binding of AtPHR1 to AtSPX2 in vitro, and consequently cannot determine the relative binding affinities of InsP₇ and InsP₈ to that complex.

Response Fig. 3. SDS-PAGE of ¹⁵N labeled and purified AtSPX2¹⁻¹⁹².

Response Fig. 4. 2D NMR spectra of four different AtSPX2 fragments indicate that the recombinantly expressed Arabidopsis SPX domain is only partially folded.

Response Fig. 5. 2D $[^{15}\text{N},^1\text{H}]$ -TROSY of $[U\text{-}^{15}\text{N}]$ -AtSPX2(1-192) in presence of 10x molar excess of InsP_6 and $[U\text{-}^{15}\text{N}]$ -AtSPX2 $^{1-192}$ (left panel) or in presence of 10x molar excess of InsP_6 and 5x molar excess of unlabeled interaction partner AtPHR1 $^{280-360}$. Buffer used contained 25 mM HEPES pH 7.0, 250 mM NaCl, 0.5 mM EDTA, 0.5 mM TCEP and 50 mM Arg/Glu mixture with 5% of D_2O measured at 15 °C. 600 MHz NMR spectrometer with cyro-probe was used. Legend: green-AtSPX2 $^{1-192}$, purple - AtSPX2 $^{1-192}$ in presence of InsP_6 , orange - AtSPX2 $^{1-192}$ in presence of InsP_6 and AtPHR1 $^{280-360}$.